# From Dictionary to Tensor: A Scalable Multi-View Subspace Clustering Framework with Triple Information Enhancement

**Zhibin Gu**[1]    **Songhe Feng**[2,3*]

[1] College of Computer and Cyber Security, Hebei Normal University, China
[2] Key Laboratory of Big Data & Artificial Intelligence in Transportation (Beijing Jiaotong University),
Ministry of Education, China
[3] School of Computer Science and Technology, Beijing Jiaotong University, China
`{guzhibin, shfeng}@bjtu.edu.cn`

## Abstract

While Tensor-based Multi-view Subspace Clustering (TMSC) has garnered significant attention for its capacity to effectively capture high-order correlations among multiple views, three notable limitations in current TMSC methods necessitate consideration: 1) high computational complexity and reliance on dictionary completeness resulting from using observed data as the dictionary, 2) inaccurate subspace representation stemming from the oversight of local geometric information and 3) under-penalization of noise-related singular values within tensor data caused by treating all singular values equally. To address these limitations, this paper presents a **S**calable TMSC framework with **T**riple inf**O**rmatio**N** **E**nhancement (**STONE**). Notably, an enhanced anchor dictionary learning mechanism has been utilized to recover the low-rank anchor structure, resulting in reduced computational complexity and increased resilience, especially in scenarios with inadequate dictionaries. Additionally, we introduce an anchor hypergraph Laplacian regularizer to preserve the inherent geometry of the data within the subspace representation. Simultaneously, an improved hyperbolic tangent function has been employed as a precise approximation for tensor rank, effectively capturing the significant variations in singular values. Extensive experiments on a variety of datasets show that the STONE outperforms SOTA approaches in both effectiveness and efficiency.

## 1   Introduction

Data clustering, a fundamental technique within the domains of machine learning and computer vision, aims to partition an unlabeled dataset into discernible subgroups characterized by substantial internal similarity [1–5]. In practical scenarios, objects are frequently characterized by a multitude of properties or data originating from various sources [6–9]. For instance, in medical analysis, imaging data from modalities such as X-ray, CT, and MRI play a crucial role in diagnosis and disease monitoring. These diverse features, representing various aspects of the same object, collectively constitute multi-view data. Multi-view clustering (MVC), which endeavors to harness the abundant information inherent in multi-view data to enhance the quality of clustering, has emerged as a highly esteemed research avenue [10–12]. Existing MVC methods can be broadly categorized into four groups based on the underlying learning mechanisms: matrix factorization-based approaches [13–15], subspace-based approaches [16–18], graph-based approaches [19–21], and kernel-based approaches [22–24]. Among these, subspace approaches are highly regarded for their straightforward implementation and excellent results.

---

*Corresponding author

38th Conference on Neural Information Processing Systems (NeurIPS 2024).

Multi-view subspace clustering is oriented towards the incorporation of diverse constraints within subspace representations to acquire a consensus one that is conducive to clustering [25–31]. For instance, Cao et al. [26] and Li et al. [30] proposed the utilization of the Hilbert-Schmidt independence criterion as a dependency measure, aiming to capture diversity and consistency from multi-view data, respectively. Pan and Kang [28] integrated a contrastive loss regularization into the consensus subspace representation, encouraging the proximity of similar samples and the separation of dissimilar ones. In addition, Huang et al. [32] facilitated the extraction of valuable consensus representation by assuming cross-view sparsity of inconsistent components in multi-view data. Nevertheless, these methods are confined to investigating only the linear affinity relationships between data pairs within individual views and do not capitalize on the higher-order correlations among data points across multiple views. This limitation results in suboptimal clustering performance. As a result, tensor-based multi-view subspace clustering methods (TMSC) have remained the focus of sustained attention in recent years. Typically, these TMSC methods consolidate various subspace representations into a 3D tensor, subsequently applying global structural constraints to uncover the complex, nonlinear relationships among data points across different views [33–39]. For example, Xie et al. [34] advanced cross view consistency exploration by employing Tensor Nuclear Norm (TNN) on the rotated tensor. Jia et al. [36] characterized the intra-view and inter-view relationships of data points by applying symmetric low-rank constraints to the frontal slices and structured sparse low-rank constraints to the horizontal slices. Furthermore, Guo et al. [38] and Sun et al. [39] introduced the logarithmic Schatten-$p$ norm and the arctan rank norm as compact surrogates for tensor rank, aimed at capturing distinctive information from tensor singular values.

Despite the noteworthy clustering quality achieved by the TMSC methods described above, there remains considerable potential for further enhancements across four critical dimensions. First, many existing methods exhibit quadratic or even cubic time and space complexities, which restricts their scalability for large-scale datasets. Second, previous techniques have utilized the given feature matrix as the dictionary for subspace recovery. However, this method requires that the feature representations to include a sufficient number of uncontaminated sampled points; otherwise, the resulting subspace representation may not accurately capture the affinity relationships among the data points. Third, traditional approaches often emphasize the low-rank structure of tensor representations to investigate high-order nonlinear correlations among data points across various views, while frequently neglecting the intricate local geometric correlations within individual view. Finally, many methods impose equal penalties on the singular values of tensor data, which may lead to excessive penalization of larger singular values while under-penalizing smaller ones, resulting in suboptimal tensor representations. This issue arises from the differing significance of singular values in tensor data, where larger singular values indicate valuable features and smaller ones are often associated with noise.

Drawing from the principles and justifications discussed previously, this paper proposes a **S**calable TMSC framework with **T**riple inf**O**rmatio**N** **E**nhancement (**STONE**). First, STONE employs an enhanced anchor dictionary representation mechanism instead of the traditional self-representation to learn a subspace representation. This approach effectively reduces computational complexity and enhances the stability and robustness of the algorithm in situations where dictionary are insufficient or corrupted. Additionally, we introduce an anchor hypergraph Laplacian regularization to guide the learning of target anchor tensor representation, facilitating the simultaneous utilization of high-order correlations among data points across views and geometric correlations among data points within each view. Furthermore, a refined hyperbolic tangent rank is developed as a non-convex low-rank regularization for tensor data, enabling the STONE model to effectively distinguish the distinct physical meanings of various singular values. Compared to existing TMSC methods, the contributions of this paper can be outlined as follows:

- We introduce an enhanced anchor dictionary representation strategy to recover the anchor subspace representation, mitigating the high computational complexity of self-representation methods and improving accuracy under dictionary under-sampling.

- We develop a refined Hyperbolic Tangent Rank (HTR) as a precise approximation to the tensor rank. In contrast to TNN, HTR allows for variable penalties on individual singular values, facilitating a thorough exploration of differences among different singular values.

- We utilize anchor hypergraphs that encode geometric manifold correlation to regularize the target tensor representation, allowing for the simultaneous utilization of high-order correlations across different views and the complex relationships within each view.

- We present an iterative optimization algorithm along with analyses of its complexity and convergence. Comprehensive experimental results demonstrate that the STONE model excels in both clustering performance and efficiency.

## 2    Theoretical Foundation

Let $\mathbf{X} = \{\mathbf{x}_1, ..., \mathbf{x}_n\} \in \mathbb{R}^{d \times n}$ denotes a dataset comprising $n$ instances, with each instance represented by a $d$-dimensional feature vector. Low-Rank Representation (LRR) [40] aims to recover a subspace representation by employing the feature matrix $\mathbf{X}$ as a dictionary, which can be mathematically described as follows:

$$\min_{\mathbf{Z},\mathbf{E}} \|\mathbf{Z}\|_* + \alpha \|\mathbf{E}\|_{2,1}, \text{s.t.} \, \mathbf{X} = \mathbf{XZ} + \mathbf{E}, \tag{1}$$

where $\mathbf{Z} \in \mathbb{R}^{n \times n}$ represents the subspace representation, which is regularized with the nuclear norm $\| \cdot \|_*$ to ensure a low-rank structure. The reconstruction error is denoted by $\mathbf{E} \in \mathbb{R}^{d \times n}$ and is constrained by the $\ell_{2,1}$-norm to promote sparsity. The parameter $\alpha$ serves as a balancing factor.

LRR has proven its effectiveness in uncovering the spatial structure of data patterns [41–43], yet it hinges on a critical requirement: the data matrix $\mathbf{X}$ must contain a sufficient number of data points sampled from the subspaces. Otherwise, a potential solution to Eq. (1) could be the identity matrix, which hinders the implementation of low-rank representation (LRR). To address this issue, Liu and Yan [44] proposed that, alongside the given data $\mathbf{X}$, there exists a set of unobserved data points $\mathbf{Y}$ in the dictionary representation, which acts as an ideal supplement to $\mathbf{X}$. This strategy is known as the latent low-rank representation model (LatLRR), which helps to mitigate the impacts of insufficient and corrupted observational data. Its mathematical definition is as follows:

$$\min_{\mathbf{Z},\mathbf{E}} \|\mathbf{Z}\|_* + \alpha \|\mathbf{E}\|_{2,1}, \text{s.t.} \, \mathbf{X} = [\mathbf{X}; \mathbf{Y}]\mathbf{Z} + \mathbf{E}, \tag{2}$$

where $\mathbf{Y} \in \mathbb{R}^{k \times n}$ represents the unobserved feature representation, which is concatenated with $\mathbf{X}$ along the columns to form a complete feature representation serving as the dictionary. For practicality, [44] relaxes Eq. (2) into the following nuclear norm minimization problem to approximate the unobserved data and learn an accurate subspace representation:

$$\min_{\mathbf{Z},\mathbf{P},\mathbf{E}} \|\mathbf{Z}\|_* + \|\mathbf{P}\|_* + \alpha \|\mathbf{E}\|_{2,1}, \text{s.t.} \, \mathbf{X} = \mathbf{XZ} + \mathbf{PX} + \mathbf{E}, \tag{3}$$

where $\mathbf{P} \in \mathbb{R}^{d \times d}$ denotes an intermediate result, which is obtained through the skinny SVD theory, which serves as a tool for feature extraction [44]. Emphasizing our focus on clustering, the subsequent discussion revolves around the subspace representation $\mathbf{Z}$, and the nuclear norm on $\mathbf{P}$ will be relaxed to the Frobenius norm—a convex surrogate for low-rank constraint that adheres to the block diagonal condition [45–47].

## 3    The Proposed Method

### 3.1    The STONE Model

Consider a dataset containing $n$ samples and $m$ views, denoted as $\{\mathbf{X}^v\}_{v=1}^m$, where $\mathbf{X}^v \in \mathbb{R}^{d_v \times n}$ represents the $v$-th view feature, and $d_v$ indicating the corresponding dimension. The objective of the TMSC method is to organize multiple view-specific subspace representations into a 3-D low-rank tensor, with the aim of unveiling higher-order correlation information spanning multiple views. Formally, the general mathematical expression of TMSC is as follows:

$$\min_{\{\mathbf{Z}^v, \mathbf{E}^v\}} \mathcal{R}(\boldsymbol{\mathcal{Z}}) + \alpha \mathcal{L}(\{\mathbf{E}^v\}) + \beta \mathcal{T}(\{\mathbf{Z}^v\})$$
$$\text{s.t.} \, \forall v, \, \mathbf{X}^v = \mathbf{X}^v \mathbf{Z}^v + \mathbf{E}^v, \, \boldsymbol{\mathcal{Z}} = \psi(\mathbf{Z}^1, ..., \mathbf{Z}^m), \tag{4}$$

where $\mathbf{Z}^v \in \mathbb{R}^{n \times n}$ represents the subspace representation of the $v$-th view, and $\boldsymbol{\mathcal{Z}} \in \mathbb{R}^{n \times m \times n}$ is a 3-D tensor formed from the collection $\{\mathbf{Z}^v\}_{v=1}^m$, with $\psi$ acting as the tensorization operator. $\mathcal{R}(\cdot)$ is used for compact approximation of the tensor rank, while $\mathcal{L}(\cdot)$ is tailored to capture noise. $\mathcal{T}(\cdot)$ represents the structured constraint applied to the subspace representation $\mathbf{Z}^v$. $\alpha$ and $\beta$ are two trade-off parameters.

Although model (4) effectively captures the high-order consistency of data points across different views, it has two notable limitations regarding its mechanism of using the observed data as a dictionary for constructing the tensor representation. First, the time and space complexity of Model (4) becomes quadratic or even cubic, which restricts its scalability to large datasets. Second, it requires the feature representation matrix $\mathbf{X}^v$ to contain a sufficient number of uncontaminated sampled data points; otherwise, the learned subspace matrix $\mathbf{Z}^v$ may manifest as the identity matrix, hindering the effectiveness of the LRR method [44].

To overcome these limitations, we introduce the Enhanced Anchor Dictionary (EAD) representation strategy for recovering anchor subspace representations. EAD first selects a set of distinctive samples from the available data to form an anchor dictionary (i.e., $\mathbf{A}^v \in \mathbb{R}^{d_v \times l}$, $l$ is the number of anchors), enabling the recovery of a subspace representation $\mathbf{Z}^v \in \mathbb{R}^{n \times l}$ that is smaller in size. This approach helps alleviate the issue of high computational complexity. Additionally, inspired by LatLRR [44], EAD integrates the observed anchors $\mathbf{A}^v$ with the unobserved sampled data $\mathbf{Y}^v$ into a comprehensive dictionary (i.e., $[\mathbf{X}^v; \mathbf{Y}^v]$), effectively mitigating problems arising from under-sampling of feature characteristics in the anchor dictionary. As a result, we formulate a TMVC framework induced by EAD as follows:

$$
\begin{aligned}
\min_{\{\mathbf{Z}^v, \mathbf{A}^v, \mathbf{P}^v, \mathbf{E}^v\}} \quad & \mathcal{R}(\boldsymbol{\mathcal{Z}}) + \alpha\mathcal{F}(\boldsymbol{\mathcal{P}}) + \beta\mathcal{L}(\{\mathbf{E}^v\}) + \gamma\mathcal{T}(\{\mathbf{Z}^v\}) \\
\text{s.t.} \, \forall v, \quad & \mathbf{X}^v = \mathbf{A}^v(\mathbf{Z}^v)^\top + \mathbf{P}^v\mathbf{X}^v + \mathbf{E}^v, (\mathbf{A}^v)^\top\mathbf{A}^v = \mathbf{I}, \\
& \boldsymbol{\mathcal{Z}} = \psi(\mathbf{Z}^1, ..., \mathbf{Z}^m), \boldsymbol{\mathcal{P}} = \psi(\mathbf{P}^1, ..., \mathbf{P}^m),
\end{aligned}
\tag{5}
$$

where $\mathbf{Z}^v \in \mathbb{R}^{n \times l}$ and $\mathbf{P}^v \in \mathbb{R}^{d^v \times d^v}$ denote the anchor subspace and projection matrix, respectively, and presented in tensor forms as $\boldsymbol{\mathcal{Z}} \in \mathbb{R}^{l \times m \times n}$ and $\boldsymbol{\mathcal{P}} \in \mathbb{R}^{d_v \times m \times d_v}$. $\mathcal{F}(\cdot)$ is a constraint on $\boldsymbol{\mathcal{P}}$. Notably, the anchor matrix $\mathbf{A}^v \in \mathbb{R}^{d_v \times l}$ is subjected to orthogonality constraints to ensure optimal distinguishability. $\alpha$, $\beta$ and $\gamma$ are three trade-off parameters.

To delve deeper into the valuable information embedded in multi-view data and refine the quality of the anchor tensor representation obtained in the Model (5), tailored constraints—including Hyperbolic Tangent Rank, Linear Weighted Frobenius norm, and Anchor Hypergraph Laplacian Regularization—are applied to $\boldsymbol{\mathcal{Z}}$, $\boldsymbol{\mathcal{P}}$, and $\mathbf{Z}^v$, respectively. These constraints are clearly defined as follows:

**Definition 1.** *For a tensor $\boldsymbol{\mathcal{Z}} \in \mathbb{R}^{n_1 \times n_2 \times n_3}$, the Hyperbolic Tangent Rank (HTR) is defined as follows:*

$$
\|\boldsymbol{\mathcal{Z}}\|_{\mathrm{HTR}} := \frac{1}{n_3} \sum_{k=1}^{n_3} \|\boldsymbol{\mathcal{Z}}_f^k\|_{\mathrm{HTR}} = \frac{1}{n_3} \sum_{k=1}^{n_3} \sum_{i=1}^{h} \left( \frac{e^{\delta\boldsymbol{\mathcal{C}}_f^k(i,i)} - e^{-\delta\boldsymbol{\mathcal{C}}_f^k(i,i)}}{e^{\delta\boldsymbol{\mathcal{C}}_f^k(i,i)} + e^{-\delta\boldsymbol{\mathcal{C}}_f^k(i,i)}} \right),
\tag{6}
$$

*where $\delta > 0$, $h = min(n_1, n_2)$. $\boldsymbol{\mathcal{Z}}_f^k$ denotes the $k$-th frontal slice of $\boldsymbol{\mathcal{Z}}$ and $\boldsymbol{\mathcal{C}}_f^k$ is the representation of the Fourier domain obtained by the tensor-SVD (i.e., $\boldsymbol{\mathcal{Z}}_f^k = \boldsymbol{\mathcal{B}}_f^k\boldsymbol{\mathcal{C}}_f^k(\boldsymbol{\mathcal{D}}_f^k)^\top$).*

**Definition 2.** *For multiple matrices $\{\mathbf{P}^v\}_{v=1}^m$, their Linearly Weighted Frobenius (LWF) norm is defined as follows:*

$$
\|\boldsymbol{\mathcal{P}}\|_{\mathrm{LWF}} := \sum_{v=1}^{m} \|\mathbf{P}^v\|_{\mathrm{LWF}} = \sum_{v=1}^{m} \xi^v\|\mathbf{P}^v\|_F,
\tag{7}
$$

*where $\|\cdot\|_F$ denotes the Frobenius norm of a matrix, and $\boldsymbol{\xi} = [\xi^1, \xi^2, ..., \xi^m]$ represents the weighted coefficient vector, with each weight empirically set to 1.*

**Definition 3.** *For the given tensor $\boldsymbol{\mathcal{Z}} \in \mathbb{R}^{l \times m \times n}$, its Anchor Hypergraph Laplacian Regularization (AHR) is defined as follows:*

$$
\|\boldsymbol{\mathcal{Z}}\|_{\mathrm{AHR}} := \sum_{v=1}^{m} \|\mathbf{Z}^v\|_{\mathrm{AHR}} = \sum_{v=1}^{m} Tr(\mathbf{Z}^v\mathbf{L}_h^v(\mathbf{Z}^v)^\top),
\tag{8}
$$

*where $\mathbf{L}_h^v$ denotes the anchor hyper-Laplacian matrix constructed based on the anchor hypergraph $\mathbf{S}_h^v(\mathbf{V}, \mathbf{Q}, \mathbf{W})$ (with $\mathbf{V}$, $\mathbf{Q}$, and $\mathbf{W}$ denoting vertices, hyperedge set, and weights, respectively). Specifically, $\mathbf{L}_h^v = \mathbf{D}_h^v - \mathbf{R}^v\mathbf{W}_e^v(\mathbf{D}_e^v)^{-1}\mathbf{R}^v$. Here, $\mathbf{D}_h^v$, $\mathbf{D}_e^v$ and $\mathbf{W}_e^v$ being degree matrices with diagonal elements as vertex degrees, hyperedge degrees and hyperedge weights, respectively. $\mathbf{R}^v$ defines vertex-hyperedge relationships, where $r^v(v, e) = 1$ if the $v$-th vertex is in the $e$-th hyperedge, otherwise 0 [48–50].*

By unifying Eqs. (5) - (8), we formulate the objective function for the STONE model as follows:

$$\min_{\{\mathbf{Z}^v, \mathbf{P}^v, \mathbf{A}^v\}, \mathbf{E},} \|\boldsymbol{\mathcal{Z}}\|_{\text{HTR}} + \alpha\|\boldsymbol{\mathcal{P}}\|_{\text{LWF}} + \beta\|\mathbf{E}\|_{2,1} + \gamma\|\boldsymbol{\mathcal{Z}}\|_{\text{AHR}}$$

$$\text{s.t. } \forall v, \ \mathbf{X}^v = \mathbf{A}^v(\mathbf{Z}^v)^\top + \mathbf{P}^v\mathbf{X}^v + \mathbf{E}^v, \mathbf{E} = [\mathbf{E}^1, \dots, \mathbf{E}^m]^\top, \tag{9}$$

$$(\mathbf{A}^v)^\top \mathbf{A}^v = \mathbf{I}, \boldsymbol{\mathcal{Z}} = \psi(\mathbf{Z}^1, \dots, \mathbf{Z}^m), \quad \boldsymbol{\mathcal{P}} = \psi(\mathbf{P}^1, \dots, \mathbf{P}^m),$$

where $\mathbf{E} = [\mathbf{E}^1; \dots; \mathbf{E}^m]^\top$ is derived by horizontally concatenating elements along the rows of $\{\mathbf{E}^v\}$. In the end, by utilizing the $k$-means clustering algorithm on the left singular vectors of the connectivity matrix $\bar{\mathbf{Z}} = \frac{1}{\sqrt{m}}[\mathbf{Z}^1, \dots, \mathbf{Z}^m] \in \mathbb{R}^{n \times lm}$, we achieve the clustering partition results [51].

**Remark 1. [Why STONE outperforms other self-representation methods?]** Unlike previous TMSC methods [38, 52], the STONE model employs the EAD strategy instead of self-representation to recover subspace representations, combining the benefits of anchor representation and latent low-rank representation (LatLRR) for the preservation of both accuracy and efficiency. Notably, illustrated in Figure 1, the efficacy of EAD stems from its thoughtful design: the utilization of the anchor representation enables the EAD model to recover the subspace representation of size $n \times l$, ensuring linear scalability for extensive

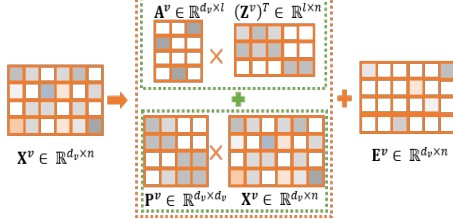

Figure 1: Schematic of Enhanced Anchor Dictionary Representation (EAD).

datasets. Additionally, the introduction of the LatLRR mechanism permits both the observed anchor vectors and the unobserved sampled data to function as dictionaries, safeguarding the recovered anchor tensor representation against deficiencies in insufficient dictionaries.

**Remark 2. [Why STONE outperforms other tensor rank methods?]** The STONE focuses on using the hyperbolic tangent tensor rank as a low-rank structural regularization constraint for tensor representation, defined as $f(x) = \frac{e^{\delta x} - e^{-\delta x}}{e^{\delta x} + e^{-\delta x}}$. Since HTR is a non-convex function with adjustable slopes, it can delve into the distinct physical meanings of different singular values in tensor data, thereby enhancing the representation capability of tensors. Analysis of Figure 2 reveals a clear superiority of the HTR in approximating tensor rank compared to TNN [34] and TLS$_p$N [38], particularly for values nearing zero and relatively large singular values. Specifically, as $x$ approaches

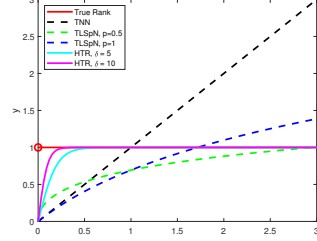

Figure 2: Tensor Rank Approximation: HTR vs. TNN and TLS$_p$N.

0, $f_{\text{HTR}}(x)$ is considerably greater than $x$ and $\log(1 + x^p)$; on the other hand, as $x$ increases, $f_{\text{HTR}}(x)$ approaches 1. The STONE method adaptively applies appropriate strong and weak penalties to both small and large singular values, preserving valuable information while also demonstrating robustness against noise. Furthermore, when $x = 0$, $f(x) = 0$, which is consistent with the true tensor rank.

## 3.2 Optimization

To tackle the objective function, we start by introducing auxiliary variables $\boldsymbol{\mathcal{S}}$ and $\{\mathbf{Q}^v\}$, which ensure that all variables in Eq. (9) become separable, as follows:

$$\min_{\{\mathbf{Z}^v, \mathbf{P}^v, \mathbf{A}^v, \mathbf{Q}^v\}, \mathbf{E}, \boldsymbol{\mathcal{S}}} \|\boldsymbol{\mathcal{S}}\|_{\text{HTR}} + \alpha\sum_{v=1}^{m}\|\mathbf{P}^v\|_F^2 + \beta\|\mathbf{E}\|_{2,1} + \gamma\sum_{v=1}^{m}\text{Tr}(\mathbf{Q}^v\mathbf{L}_h^v(\mathbf{Q}^v)^\top) + \frac{\theta}{2}\|\boldsymbol{\mathcal{Z}} - \boldsymbol{\mathcal{S}} + \frac{\boldsymbol{\mathcal{Y}}}{\theta}\|_F^2$$

$$+ \frac{\epsilon_1}{2}\sum_{v=1}^{m}\|\mathbf{X}^v - \mathbf{A}^v(\mathbf{Z}^v)^\top - \mathbf{P}^v\mathbf{X}^v - \mathbf{E}^v + \frac{\mathbf{H}_1^v}{\epsilon_1}\|_F^2 + \frac{\epsilon_2}{2}\sum_{v=1}^{m}\|\mathbf{Z}^v - \mathbf{Q}^v + \frac{\mathbf{H}_2^v}{\epsilon_2}\|_F^2, \tag{10}$$

where $\boldsymbol{\mathcal{Y}}$, $\{\mathbf{H}_1^v\}$ and $\{\mathbf{H}_2^v\}$ are the Lagrange multipliers, and $\theta$, $\epsilon_1$ and $\epsilon_2$ signifies the penalty coefficients. Then, the optimization of the STONE objective function can be streamlined into six sub-problems labeled as $\{\mathbf{Z}^v\}$, $\{\mathbf{A}^v\}$, $\{\mathbf{P}^v\}$, $\{\mathbf{Q}^v\}$, $\mathbf{E}$ and $\boldsymbol{\mathcal{S}}$ for individual optimization. Given space limitations, the comprehensive optimization procedures and pseudocode are outlined in the A.1 of the supplementary materials.

### 3.3 Convergence Analysis

The validation presented in Theorem 1 establishes the reliability of the optimization algorithm's convergence, while Appendix A.2 of the supplementary materials offers an in-depth exploration of the underlying details.

**Theorem 1.** *The sequence generated by the employed optimization algorithm, denoted as $\mathcal{G}_t = \{\mathbf{Z}_t^v, \mathbf{E}_t^v, \mathbf{P}_t^v, \mathbf{A}_t^v, \mathbf{H}_{1t}^v, \mathbf{H}_{2t}^v, \mathbf{Q}_t^v, \mathcal{S}_t\}_{t=1}^{\infty}$, adheres to the following two fundamental principles:*

- *The set $\{\mathcal{G}_t\}_{t=1}^{\infty}$ is bounded;*

- *Any accumulation point of the sequence $\{\mathcal{G}_t\}_{t=1}^{\infty}$ is a KKT point of Eq.(10).*

### 3.4 Complexity Analysis

The computational requirements of STONE are split into two primary areas: optimizing variables and performing clustering. At the outset, the process involves updating several key variables-$\mathbf{L}_h^v$, $\mathbf{Z}^v$, $\mathbf{E}^v$, $\mathbf{P}^v$, $\mathbf{A}^v$, $\mathbf{Q}^v$, $\mathcal{S}$ -with their respective time complexities being $\mathcal{O}(l^2 m \log(l))$, $\mathcal{O}(nl^2 + nld^v)$, $\mathcal{O}(nd^v)$, $\mathcal{O}(n(d^v)^2 + nld^v)$, $\mathcal{O}(nld^v + l^2 d^v)$, $\mathcal{O}(nl^2)$, $\mathcal{O}(mnl \log(mn) + nm^2 l)$. In the following phase, the computational complexity is given by $O(nlm + nd_{\max})$. This indicates a direct proportionality to the sample size $n$. Furthermore, the memory complexity of the STONE model, expressed as $O(nlm + nd_{\max})$, also maintains a linear growth pattern with respect to $n$.

### 3.5 Comparison with Previous Studies

In recent years, various tensor-based multi-view clustering algorithms, such as MVSC-TLRR [36], TLS$_p$NM-MSC [38], SSG-TAR [39], NOODLE [53], ASR-ETR [33], and EDISON [54], have been proposed to explore high-order correlations among views by pursuing a global low-rank structure in tensor representations. However, our STONE model significantly differs from these methods. For instance, unlike MVSC-TLRR, TLS$_p$NM-MSC, SSG-TAR and NOODLE, our STONE model differs by enhancing computational efficiency through the construction of anchor subspace representations rather than relying on traditional subspace representations. Moreover, unlike the ASR-ETR, which lowers computational complexity through anchor dictionary representation, our STONE model builds on this by using the EAD strategy to address challenges related to insufficient data sampling. Additionally, STONE employs anchor hypergraph Laplacian regularization rather than anchor Laplacian regularization in ASR-ETR, which further enhances the accuracy of subspace representations. In contrast to the EDISON, designed for incomplete multi-view data, our approach not only has differences in dictionary representations due to variations in data completeness, but also employs distinct non-convex functions to regularize the singular values of tensor data during the recovery of compact tensor representations.

## 4 Experiment

In this section, we present comprehensive experiments to evaluate the performance of the STONE model. Due to space constraints, a portion of the experiments is presented here, with additional experiments detailed in the Appendix A.3 of the supplementary materials.

### 4.1 Experimental Setup

**Datasets:** For the clustering experiments, we employ eight datasets: NGs, BBCSport, HW, Scene15, MSRCV1, Caltech101-all, ALOI-100, and CIFAR10. More detailed descriptions of these datasets can be found in Table 1.

**Baselines:** Ten SOTA methods, including eight shallow-based models and two deep learning models: SMVSC (2021) [55], SFMC (2022) [56], GMC (2019) [57], MSC$^2$D (2023) [58], MVCtopl (2022) [59], MVSCTM (2022) [60], ETLMC (2019) [61],

Table 1: Overview of Statistical Features for Eight Datasets.

| Datasets | Type | Samples | Clusters | Views |
|---|---|---|---|---|
| NGs | Text | 500 | 5 | 3 |
| BBCSport | Text | 544 | 5 | 2 |
| HW | Digit | 2000 | 10 | 2 |
| Scene15 | Scene | 4485 | 15 | 3 |
| MSRCV1 | Object | 210 | 7 | 5 |
| Caltech101-all | Object | 9144 | 102 | 6 |
| ALOI-100 | Object | 10800 | 100 | 4 |
| CIFAR10 | Object | 50000 | 10 | 4 |

Table 2: Clustering Performance Comparison Across Eight Datasets (Mean ± Standard Deviation).

| Dataset | Metric | SC-best | SMVSC | SFMC | GMC | MSC²D | MVCtopl | MVSCTM | ETLMSC | TBGL | MFLVC | GCFAgg | STONE |
|---|---|---|---|---|---|---|---|---|---|---|---|---|---|
| **NGs** | ACC | 0.259±0.001 | 0.710±0.000 | 0.218±0.000 | 0.982±0.000 | 0.977±0.003 | 0.440±0.000 | 0.264±0.000 | 0.679±0.001 | 0.236±0.000 | 0.710±0.000 | 0.650±0.000 | **1.000±0.000** |
| | NMI | 0.018±0.000 | 0.564±0.000 | 0.021±0.000 | 0.939±0.000 | 0.927±0.010 | 0.351±0.000 | 0.071±0.000 | 0.604±0.002 | 0.035±0.000 | 0.567±0.000 | 0.506±0.000 | **1.000±0.000** |
| | PUR | 0.259±0.001 | 0.712±0.000 | 0.220±0.000 | 0.982±0.000 | 0.977±0.003 | 0.500±0.000 | 0.266±0.000 | 0.707±0.001 | 0.240±0.000 | 0.736±0.000 | 0.680±0.000 | **1.000±0.000** |
| | F-score | 0.221±0.000 | 0.628±0.000 | 0.329±0.000 | 0.964±0.000 | 0.955±0.006 | 0.432±0.000 | 0.334±0.000 | 0.647±0.001 | 0.327±0.000 | 0.632±0.000 | 0.569±0.000 | **1.000±0.000** |
| | ARI | 0.005±0.000 | 0.520±0.000 | 0.001±0.000 | 0.955±0.000 | 0.944±0.008 | 0.205±0.000 | 0.015±0.000 | 0.551±0.01 | 0.002±0.000 | 0.534±0.000 | 0.461±0.000 | **1.000±0.000** |
| **BBCSport** | ACC | 0.504±0.004 | 0.507±0.000 | 0.366±0.000 | 0.807±0.000 | 0.606±0.001 | 0.752±0.000 | 0.421±0.000 | 0.961±0.000 | 0.548±0.000 | 0.643±0.000 | 0.638±0.000 | **1.000±0.000** |
| | NMI | 0.210±0.002 | 0.210±0.000 | 0.021±0.000 | 0.723±0.000 | 0.479±0.018 | 0.601±0.000 | 0.201±0.000 | 0.890±0.000 | 0.277±0.000 | 0.453±0.000 | 0.393±0.000 | **1.000±0.000** |
| | PUR | 0.553±0.004 | 0.535±0.000 | 0.369±0.000 | 0.844±0.000 | 0.642±0.007 | 0.789±0.000 | 0.461±0.000 | 0.961±0.000 | 0.550±0.000 | 0.684±0.000 | 0.638±0.000 | **1.000±0.000** |
| | F-score | 0.414±0.002 | 0.368±0.000 | 0.385±0.000 | 0.794±0.000 | 0.581±0.008 | 0.689±0.000 | 0.445±0.000 | 0.938±0.000 | 0.473±0.000 | 0.510±0.000 | 0.482±0.000 | **1.000±0.000** |
| | ARI | 0.159±0.005 | 0.171±0.000 | 0.004±0.000 | 0.722±0.000 | 0.373±0.014 | 0.577±0.000 | 0.125±0.000 | 0.919±0.000 | 0.188±0.000 | 0.375±0.000 | 0.339±0.000 | **1.000±0.000** |
| **HW** | ACC | 0.739±0.000 | 0.821±0.000 | 0.859±0.000 | 0.882±0.000 | 0.879±0.010 | 0.631±0.000 | 0.653±0.000 | 0.998±0.000 | 0.863±0.000 | 0.871±0.000 | 0.829±0.000 | **1.000±0.000** |
| | NMI | 0.699±0.000 | 0.789±0.000 | 0.900±0.000 | 0.893±0.000 | 0.892±0.011 | 0.676±0.000 | 0.691±0.000 | 0.995±0.000 | 0.890±0.000 | 0.883±0.000 | 0.791±0.000 | **1.000±0.000** |
| | PUR | 0.739±0.000 | 0.821±0.000 | 0.883±0.000 | 0.882±0.000 | 0.879±0.010 | 0.674±0.000 | 0.696±0.000 | 0.998±0.000 | 0.881±0.000 | 0.871±0.000 | 0.829±0.000 | **1.000±0.000** |
| | F-score | 0.653±0.000 | 0.752±0.000 | 0.855±0.000 | 0.865±0.000 | 0.860±0.018 | 0.595±0.000 | 0.613±0.000 | 0.996±0.000 | 0.854±0.000 | 0.850±0.000 | 0.744±0.000 | **1.000±0.000** |
| | ARI | 0.612±0.000 | 0.723±0.000 | 0.838±0.000 | 0.850±0.000 | 0.844±0.021 | 0.542±0.000 | 0.563±0.000 | 0.996±0.000 | 0.836±0.000 | 0.833±0.000 | 0.715±0.000 | **1.000±0.000** |
| **Scene15** | ACC | 0.229±0.005 | 0.336±0.000 | 0.092±0.000 | 0.140±0.000 | 0.274±0.059 | 0.631±0.000 | 0.196±0.000 | 0.847±0.030 | 0.298±0.000 | 0.328±0.000 | 0.341±0.000 | **0.977±0.000** |
| | NMI | 0.204±0.002 | 0.323±0.000 | 0.000±0.000 | 0.058±0.000 | 0.218±0.072 | 0.676±0.000 | 0.160±0.000 | 0.867±0.016 | 0.257±0.000 | 0.344±0.000 | 0.359±0.000 | **0.962±0.000** |
| | PUR | 0.288±0.003 | 0.348±0.000 | 0.092±0.000 | 0.146±0.000 | 0.290±0.056 | 0.674±0.000 | 0.232±0.000 | 0.886±0.017 | 0.302±0.000 | 0.339±0.000 | 0.383±0.000 | **0.977±0.000** |
| | F-score | 0.151±0.003 | 0.242±0.000 | 0.129±0.000 | 0.132±0.000 | 0.172±0.041 | 0.595±0.000 | 0.133±0.000 | 0.831±0.033 | 0.199±0.000 | 0.245±0.000 | 0.242±0.000 | **0.958±0.000** |
| | ARI | 0.085±0.003 | 0.170±0.000 | 0.000±0.000 | 0.004±0.000 | 0.060±0.054 | 0.542±0.000 | 0.015±0.000 | 0.818±0.035 | 0.102±0.000 | 0.183±0.000 | 0.187±0.000 | **0.955±0.000** |
| **MSRCV1** | ACC | 0.643±0.000 | 0.819±0.000 | 0.810±0.000 | 0.748±0.000 | 0.846±0.052 | 0.367±0.000 | 0.376±0.000 | 0.962±0.000 | **1.000±0.000** | 0.414±0.000 | 0.543±0.000 | **1.000±0.000** |
| | NMI | 0.555±0.000 | 0.718±0.000 | 0.721±0.000 | 0.742±0.000 | 0.780±0.017 | 0.287±0.000 | 0.296±0.000 | 0.937±0.000 | **1.000±0.000** | 0.387±0.000 | 0.496±0.000 | **1.000±0.000** |
| | PUR | 0.700±0.000 | 0.819±0.000 | 0.810±0.000 | 0.790±0.000 | 0.855±0.032 | 0.400±0.000 | 0.410±0.000 | 0.962±0.000 | **1.000±0.000** | 0.419±0.000 | 0.557±0.000 | **1.000±0.000** |
| | F-score | 0.531±0.000 | 0.699±0.000 | 0.714±0.000 | 0.697±0.000 | 0.754±0.026 | 0.295±0.000 | 0.295±0.000 | 0.928±0.000 | **1.000±0.000** | 0.372±0.000 | 0.433±0.000 | **1.000±0.000** |
| | ARI | 0.452±0.000 | 0.649±0.000 | 0.663±0.000 | 0.640±0.000 | 0.712±0.032 | 0.155±0.000 | 0.157±0.000 | 0.917±0.000 | **1.000±0.000** | 0.244±0.000 | 0.346±0.000 | **1.000±0.000** |
| **ALOI-100** | ACC | 0.651±0.011 | 0.351±0.000 | 0.680±0.000 | 0.721±0.000 | 0.689±0.035 | 0.441±0.000 | 0.443±0.000 | 0.767±0.000 | 0.694±0.000 | 0.274±0.000 | 0.807±0.000 | **0.814±0.000** |
| | NMI | 0.802±0.003 | 0.613±0.000 | 0.702±0.000 | 0.744±0.000 | 0.732±0.026 | 0.652±0.000 | 0.619±0.000 | 0.862±0.000 | 0.728±0.000 | 0.687±0.000 | 0.908±0.000 | **0.909±0.000** |
| | PUR | 0.677±0.010 | 0.359±0.000 | 0.691±0.000 | 0.731±0.000 | 0.707±0.028 | 0.514±0.000 | 0.509±0.000 | 0.789±0.000 | 0.709±0.000 | 0.274±0.000 | 0.824±0.000 | **0.849±0.000** |
| | F-score | 0.553±0.009 | 0.217±0.000 | 0.129±0.000 | 0.173±0.000 | 0.166±0.034 | 0.187±0.000 | 0.111±0.000 | 0.687±0.000 | 0.162±0.000 | 0.228±0.000 | **0.760±0.000** | 0.736±0.000 |
| | ARI | 0.548±0.009 | 0.206±0.000 | 0.114±0.000 | 0.158±0.000 | 0.151±0.035 | 0.173±0.000 | 0.095±0.000 | 0.684±0.000 | 0.147±0.000 | 0.215±0.000 | **0.757±0.000** | 0.733±0.000 |
| **Caltech101-all** | ACC | 0.193±0.004 | 0.307±0.000 | 0.241±0.000 | 0.195±0.000 | 0.225±0.048 | 0.121±0.000 | 0.122±0.000 | OM | OM | 0.217±0.000 | 0.197±0.000 | **0.650±0.000** |
| | NMI | 0.403±0.004 | 0.397±0.000 | 0.206±0.000 | 0.238±0.000 | 0.224±0.051 | 0.183±0.000 | 0.181±0.000 | OM | OM | 0.275±0.000 | 0.431±0.000 | **0.862±0.000** |
| | PUR | 0.404±0.007 | 0.370±0.000 | 0.290±0.000 | 0.301±0.000 | 0.296±0.052 | 0.208±0.000 | 0.213±0.000 | OM | OM | 0.287±0.000 | 0.379±0.000 | **0.855±0.000** |
| | F-score | 0.147±0.007 | 0.266±0.000 | 0.054±0.000 | 0.050±0.000 | 0.057±0.005 | 0.050±0.000 | 0.048±0.000 | OM | OM | 0.172±0.000 | 0.229±0.000 | **0.528±0.000** |
| | ARI | 0.132±0.007 | 0.235±0.000 | 0.000±0.000 | -0.004±0.000 | 0.003±0.006 | -0.002±0.000 | -0.005±0.000 | OM | OM | 0.134±0.000 | 0.216±0.000 | **0.519±0.000** |
| **CIFAR10** | ACC | 0.907±0.000 | 0.990±0.000 | 0.988±0.000 | OM | OM | OM | OM | OM | OM | 0.993±0.000 | 0.989±0.000 | **0.994±0.000** |
| | NMI | 0.805±0.000 | 0.973±0.000 | 0.969±0.000 | OM | OM | OM | OM | OM | OM | 0.980±0.000 | 0.974±0.000 | **0.984±0.000** |
| | PUR | 0.907±0.000 | 0.990±0.000 | 0.988±0.000 | OM | OM | OM | OM | OM | OM | 0.993±0.000 | 0.989±0.000 | **0.994±0.000** |
| | F-score | 0.827±0.000 | 0.980±0.000 | 0.977±0.000 | OM | OM | OM | OM | OM | OM | 0.986±0.000 | 0.979±0.000 | **0.989±0.000** |
| | ARI | 0.807±0.000 | 0.978±0.000 | 0.975±0.000 | OM | OM | OM | OM | OM | OM | 0.985±0.000 | 0.976±0.000 | **0.988±0.000** |

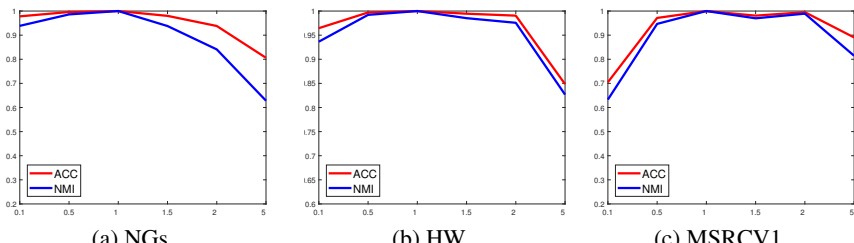

(a) NGs     (b) HW     (c) MSRCV1

Figure 3: Impact of Parameter $\delta$ on the STONE Model.

TBGL (2023) [62], MFLVC (2022) [63], GCFAgg (2023) [64], along with spectral clustering with the best view (SC-best) [65], are used for comparison.

**Evaluation Metrics:** To provide a comprehensive evaluation of clustering quality, we employ five metrics, namely ACC, NMI, PUR, F-score, and ARI. Better clustering quality is indicated by higher values of these metrics.

**Implementation Overview:** For the comparative methods, the parameters are fine-tuned in accordance with the instructions presented in the respective literature, and the optimal outcomes are reported. For the STONE model, there are five parameters that necessitate adjustment. To be specific, the intrinsic parameter $\delta$ and the number of anchor points $c$ are tuned individually within the ranges [0.1, 0.5, 1, 1.5, 5] and [c, 2c, ..., 7c], respectively. The three balancing parameters $\alpha$, $\beta$, and $\gamma$ are finely tuned within the range [1e-5,1e-5,..., 1e+1] using a grid search strategy. To maintain rigor, we perform each experiment a total of 10 times, and we present both the mean results and the standard deviations for comparison. The experimental procedures for the shallow learning model are implemented using MATLAB 2018a on a computer featuring a 3.70GHz i9-10900k CPU and 64GB RAM. Conversely, the deep learning model experiments are facilitated by PyTorch 1.12, deployed on an RTX 4060 GPU.

## 4.2 Comparison of Clustering Performance and Efficiency

The clustering performance and computational efficiency of the proposed STONE model are demonstrated separately in this subsection.

**Performance Assessment:** To validate the effectiveness of our STONE method, we evaluate its clustering performance on eight datasets and compared it with ten SOTA methods across five metrics.

Table 3: Efficiency Comparison of Different Methods on Datasets with over 4,000 Samples.

| Datasets | SMVSC | SFMC | GMC | MSC²D | MVCtopl | MVSCTM | ETLMSC | TBGL | MFLVC | GCFAgg | STONE |
|---|---|---|---|---|---|---|---|---|---|---|---|
| Scene15 | 19.79 | 23.91 | 57.14 | 174.1 | 131.91 | 482.22 | 639.95 | 1279.9 | 107.59 | 145.65 | **6.52** |
| ALOI-100 | 197.76 | 148.75 | 440.36 | 1013.9 | 7067.8 | 2064.7 | 4257.3 | 26109 | 659.69 | 400.85 | **66.57** |
| Caltech101-all | 247.29 | **165.36** | 398.94 | 856.3 | 5704.9 | 1169.2 | OM | OM | 1212.5 | 589.01 | 285.11 |
| CIFAR10 | 867.26 | 4251.3 | OM | OM | OM | OM | OM | OM | 1257.08 | 1978.88 | **860.8** |

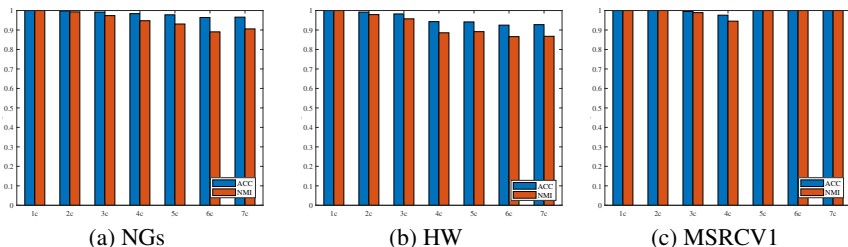

(a) NGs  (b) HW  (c) MSRCV1

Figure 4: The Influence of Anchor Quantity on STONE Model Performance.

The results are summarized in Table 2, where the highest and second-highest values are marked with **bold** and underlined, respectively. The acronym 'OM' signifies occurrences of out-of-memory errors. From Table 2, we can draw the following three findings:

1) Our STONE model exhibits excellent clustering performance across all datasets, significantly surpassing competitors in certain scenarios. For instance, on the Scene15 dataset, STONE outperforms the second-ranked method, ETMSLC, across five performance metrics (ACC, NMI, PUR, F-score, and ARI) with improvements of **13%**, **9.5%**, **9.1%**, **12.7%** and **13.7%**, respectively. Moreover, STONE demonstrates **ideal clustering performance on the NGs, BBCSport, HW, and MSRCV1** datasets. These results indicate that the STONE model effectively uncovers higher-order correlations among multiple views as well as the geometric manifold information within each view, contributing to improved clustering outcomes.

2) Methods based on tensor constraints often outperform those based on matrix constraints in terms of clustering performance. This enhancement is primarily attributed to the ability of tensor-based approaches to impose low-rank constraints at the tensor level, effectively capturing the inherent high-order correlations in multi-view data. In contrast, matrix-based methods generally focus only on linear correlations within individual views.

3) In comparison to prominent deep learning models, such as MFLVC [63] and GCFAgg [64], the STONE method demonstrates superior performance in most scenarios. This suggests that shallow learning models can still produce more effective clustering results than deep learning methods in multi-view tasks by cleverly extracting the rich information contained within multi-view data.

**Efficiency Assessment:** To demonstrate the efficiency of the STONE method, we record its running time on datasets containing over 4000 instances and compared it with other benchmark methods. The comparison results are summarized in Table 3. Notably, STONE exhibits significant efficiency in this comparison. For instance, on the ALOI-100 dataset, our method runs in 66.57 seconds, whereas the anchor tensor-induced model TBGL takes over 26000 seconds, which is considerably longer than the STONE model. The STONE method achieves higher efficiency due to its innovative combination of anchor point dictionary representation learning and anchor hypergraph Laplacian regularization. This approach selectively incorporates a small subset of the most discriminative anchor points, ensuring faster computational efficiency while preserving precise clustering performance.

## 4.3 Parameters Analysis

In the STONE model, there are five parameters, including the built-in parameter $\delta$, the number of anchors $l$, and three balancing parameters $\alpha$, $\beta$, and $\gamma$. This subsection investigates the impact of these parameters on the STONE model. Specifically, the parameters $\delta$ and $l$ are treated as independent variables for individual tuning, while the balancing parameters are adjusted pairwise using a grid search strategy.

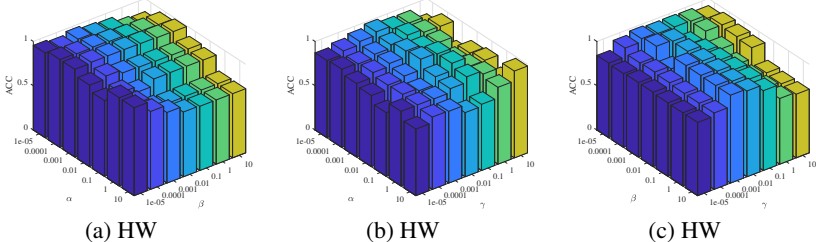

| (a) HW | (b) HW | (c) HW |

Figure 5: Sensitivity Analysis of the STONE Model to the Balance Parameters $\alpha$, $\beta$ and $\gamma$.

**Impact of the Built-in Parameter $\delta$:** HTR is utilized as a non-convex penalty term for the singular values of tensor data, dynamically controlling the degree of shrinkage applied to different singular values by adjusting the parameteras $\delta$. We explore the impact on clustering results for datasets NGs, HW, and MSRCV1 by adjusting the parameter $\delta$ across the values [0.1, 0.5, 1, 1.5, 2, 5]. This variation enabled us to assess its impact on the clustering outcomes, with the results detailed in Figure 3. Clearly, alterations in the value of $\delta$ lead to fluctuations in the clustering results, driven by the varying contraction degree of $\delta$ across different singular values.

**Influence of the Number of Anchors:** In this subsection, we study how the number of anchors affects the performance of STONE, with anchor counts ranging from [$c$, $7c$] and a step size of $c$. As illustrated in Figure 4, the clustering performance of STONE shows a fluctuating pattern as the number of anchors varies. Interestingly, the performance curve does not monotonically increase with the number of anchors, which means that choosing a smaller number of discriminant anchors is preferable to choosing a larger number of non-discriminant anchors. In addition, the best clustering quality can be obtained by using $c$ or $2c$ anchor points in STONE, which shows that the coordination between the EAD and the AHR improves the discrimination of the anchor points.

**Sensitivity Analysis of Balancing Parameters:** To assess the importance of the balancing parameters $\alpha$, $\beta$, and $\gamma$ in the STONE model, we implement a grid search strategy across the range of [1e-5, 1e+1] to optimize these parameters. Figure 5 demonstrates how the model's performance changes with various combinations of these parameters, highlighting fluctuations in clustering performance based on the chosen values. Notably, optimal performance can be achieved through careful tuning, suggesting that the modules within the STONE model can effectively coordinate their importance to extract valuable information, thereby enhancing clustering performance.

### 4.4 Convergence Behavior

This subsection provides an experimental validation of the convergence of the STONE model, utilizing two key metrics: reconstruction error (RE), defined as $\text{RE} = \sum_{v=1}^{m} \|\mathbf{X}^v - \mathbf{A}^v (\mathbf{Z}^v)^\top - \mathbf{P}^v \mathbf{X}^v - \mathbf{E}^v\|_\infty$ and matching error (ME), represented as $\text{ME} = \|\boldsymbol{\mathcal{Z}} - \boldsymbol{\mathcal{S}}\|_\infty$. The iterative trends observed on the NGs, HW, and MSRCV1 datasets, as depicted in Figure 6, demonstrate that both RE and ME exhibit rapid convergence to 0 within 15 iterations, followed by stabilization. This outcome substantiates the robust convergence properties of the STONE method.

### 4.5 Ablation Study

Comprehensive ablation experiments are carried out in this subsection to systematically assess the contributions of various modules within the STONE model. Here, we assign the values of the balancing parametersset $\alpha$, $\beta$, and $\gamma$—governing the loss terms $\mathcal{L}_{EAD}$, $\mathcal{L}_{RE}$, and $\mathcal{L}_{AHR}$, respectively—to 0, essentially isolating and removing each loss term separately from the STONE model. The experimental results for the

Table 4: Analysis of STONE Model Ablation.

| Datasets | | | NGs | | MSRCV1 | | HW | |
|---|---|---|---|---|---|---|---|---|
| $\mathcal{L}_{EAD}$ | $\mathcal{L}_{RE}$ | $\mathcal{L}_{AHR}$ | ACC | NMI | ACC | NMI | ACC | NMI |
| ✔ | | | 0.438 | 0.291 | 0.148 | 0.030 | 0.988 | 0.974 |
| | ✔ | | 0.208 | 0.012 | 0.205 | 0.033 | 0.977 | 0.951 |
| | | ✔ | 0.596 | 0.473 | 0.148 | 0.030 | 0.988 | 0.974 |
| ✔ | ✔ | | 0.960 | 0.897 | 0.976 | 0.946 | 0.881 | 0.841 |
| ✔ | | ✔ | 0.534 | 0.397 | 0.786 | 0.631 | 0.983 | 0.971 |
| | ✔ | ✔ | 0.458 | 0.311 | 0.571 | 0.384 | 0.854 | 0.841 |
| ✔ | ✔ | ✔ | **1.000** | **1.000** | **1.000** | **1.000** | **1.000** | **1.000** |

NGs, MSRCV1 and HW datasets are presented in Table 4, with checkmarks denoting the consideration of the corresponding loss. The best-performing results are indicated in **bold**. Table 4 reveals

Table 5: Comparison of STONE and STONE-v1 across Different Datasets.

| Datasets | NGs | BBCSport | HW | Scene15 | MSRCV1 | ALOI-100 | Cal101-all | CIFAR10 |
|---|---|---|---|---|---|---|---|---|
| STONE-v1 | 0.379±0.000 | 0.648±0.000 | 0.740±0.000 | 0.629±0.000 | 0.469±0.000 | 0.600±0.000 | 0.319±0.000 | 0.501±0.000 |
| STONE | 1.000±0.000 | 1.000±0.000 | 1.000±0.000 | 0.977±0.000 | 1.000±0.000 | 0.814±0.000 | 0.650±0.000 | 0.994±0.000 |

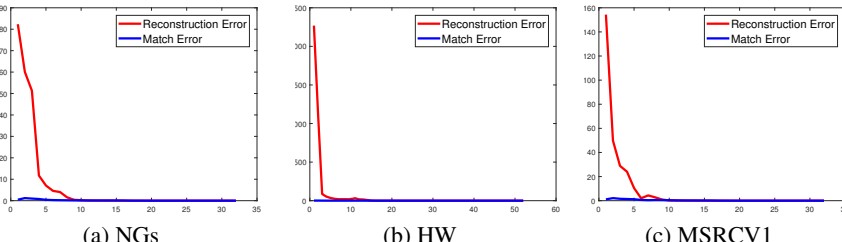

(a) NGs         (b) HW         (c) MSRCV1

Figure 6: Convergence Curves of STONE on Three Datasets.

that the clustering performance of degraded models, achieved by removing one or two submodules from the STONE model, is notably inferior to that of the complete STONE model. This emphasizes the successful collaboration of $\mathcal{L}_{EAD}$, $\mathcal{L}_{RE}$, and $\mathcal{L}_{AHR}$ within the STONE framework, allowing them to synergistically exploit the abundant information embedded in multi-view data and attain commendable clustering performance. Additionally, HTR is a novel tensor low-rank constraint in our STONE model, aimed at capturing high-order correlations and managing variations in tensor singular values. To assess its impact, we conduct an ablation study comparing the original STONE model with a version that excludes the HTR module (referred to as STONE-v1). Table 5 shows the clustering ACC across different datasets, revealing a drop in performance on all datasets when the HTR module is removed. This suggests that the integration of HTR enhances the exploration of high-order correlations, thereby improving the quality of data partitioning.

## 5 Conclusion

This paper introduces a novel tensor-based multi-view subspace clustering framework that integrates triple information enhancement from dictionary to tensor representation. Through the design of the enhanced anchor dictionary representation, hyperbolic tangent rank, and anchored hypergraph Laplacian regularization, our model extensively investigates valuable insights within multi-view data. Experimental results demonstrate that the STONE model outperforms SOTA models on eight datasets in terms of both effectiveness and efficiency.

## Acknowledgements

This work was supported by the Beijing Natural Science Foundation (No. 4242046) and the Fundamental Research Funds for the Central Universities (No. 2022JBZY019).

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

## A  Appendix / supplemental material

This supplementary material offers a comprehensive elaboration on the optimization steps presented in the main manuscript, as well as the validation of the theorems. We also include additional experimental results.

### A.1  Optimization of the Algorithm

As detailed in the main text, auxiliary variables $\boldsymbol{\mathcal{S}}$ and $\{\mathbf{Q}^v\}$ are introduced to facilitate the independent optimization of each variable.

$$
\begin{aligned}
\min_{\{\mathbf{Z}^v, \mathbf{P}^v, \mathbf{A}^v, \mathbf{Q}^v\}, \mathbf{E}, \boldsymbol{\mathcal{S}}} & \|\boldsymbol{\mathcal{S}}\|_{\text{HTR}} + \frac{\alpha}{2} \sum_{v=1}^{m} \|\mathbf{P}^v\|_F^2 + \beta \|\mathbf{E}\|_{2,1} + \gamma \sum_{v=1}^{m} \text{Tr}(\mathbf{Q}^v \mathbf{L}_h^v (\mathbf{Q}^v)^\top) \\
& + \frac{\theta}{2} \|\boldsymbol{\mathcal{Z}} - \boldsymbol{\mathcal{S}} + \frac{\boldsymbol{\mathcal{Y}}}{\theta}\|_F^2 + \frac{\epsilon_1}{2} \sum_{v=1}^{m} \|\mathbf{X}^v - \mathbf{A}^v(\mathbf{Z}^v)^\top - \mathbf{P}^v \mathbf{X}^v - \mathbf{E}^v + \frac{\mathbf{Y}_1^v}{\epsilon_1}\|_F^2 \\
& + \frac{\epsilon_2}{2} \sum_{v=1}^{m} \|\mathbf{Z}^v - \mathbf{Q}^v + \frac{\mathbf{Y}_2^v}{\epsilon_2}\|_F^2,
\end{aligned}
\tag{11}
$$

Next, we utilize the ADMM algorithm [66] to individually optimize each variable as follows:

$\{\mathbf{Z}^v\}$ **Subproblem**: Under the assumption that all other variables remain constant while $\mathbf{Z}^v$ varies, the optimization in Eq (11) simplifies to a single-variable optimization problem. To find the optimal

solution, we take the partial derivative of Eq (11) with respect to $\mathbf{Z}^v$ and set it to zero, resulting in the following solution:

$$\begin{aligned}
\mathbf{Z}^v = (\theta \boldsymbol{\mathcal{S}}^v - \boldsymbol{\mathcal{Y}}^v + \epsilon_1(\mathbf{X}^v)^\top \mathbf{A}^v - \epsilon_1(\mathbf{E}^v)^\top \mathbf{A}^v - \epsilon_1(\mathbf{X}^v)^\top(\mathbf{Q}^v)^\top \mathbf{A}^v \\
+ \epsilon_1(\mathbf{H}_1^v)^\top \mathbf{A}^v + \epsilon_2 \mathbf{Q}^v - \mathbf{H}_2^v) \times [(\theta + \epsilon_2)\mathbf{I} + \epsilon_1(\mathbf{A}^v)^\top \mathbf{A}^v]^{-1}.
\end{aligned} \tag{12}$$

$\{\mathbf{P}^v\}$ **Subproblem:** In this case, we treat only $\mathbf{P}^v$ as a variable. To find the optimal solution, we set the first derivative of Eq. (11) with respect to $\mathbf{P}^v$ to zero, resulting in the following expression:

$$\begin{aligned}
\mathbf{P}^v = &(\epsilon_1 \mathbf{X}^v - \epsilon_1 \mathbf{A}^v(\mathbf{Z}^v)^\top - \epsilon_1 \mathbf{E}^v + \mathbf{H}_1^v) \\
&\times (\mathbf{X}^v)^\top(\alpha \mathbf{I} + \epsilon_1 \mathbf{X}^v(\mathbf{X}^v)^\top)^{-1}.
\end{aligned} \tag{13}$$

**E Subproblem:** In a similar vein, assuming that all variables in Eq (11) are constant except for $\mathbf{E}$, we can reframe the optimization problem for $\mathbf{E}$ as follows:

$$\min_{\mathbf{E}} \frac{\beta}{\epsilon_1} \|\mathbf{E}\|_{2,1} + \frac{1}{2} \|\mathbf{E} - \mathbf{R}\|_F^2. \tag{14}$$

where $\mathbf{R}$ is constructed by horizontally stacking $\mathbf{X}^v - \mathbf{A}^v(\mathbf{Z}^v)^\top - \mathbf{P}^v \mathbf{X}^v + \frac{\mathbf{H}_1^v}{\epsilon_1}$. The optimal solution for $\mathbf{E}$ can then be derived by solving Eq (14) as follows:

$$\mathbf{E}_{:,i}^* = \begin{cases} \frac{\|\mathbf{R}_{:,i}\|_2 - \frac{\beta}{\epsilon_1}}{\|\mathbf{R}_{:,i}\|_2} \mathbf{R}_{:,i}, & \|\mathbf{R}_{:,i}\|_2 > \frac{\beta}{\epsilon_1} \\ 0, & otherwise \end{cases} \tag{15}$$

$\{\mathbf{A}^v\}$ **Subproblem:** When the other variables are held constant and $\mathbf{A}^v$ is treated as the variable, the optimization problem for $\mathbf{A}^v$ can be reformulated as follows:

$$\max_{\mathbf{A}^v} \mathrm{Tr}((\mathbf{A}^v)^\top \mathbf{K}), \quad \text{s.t.} \, (\mathbf{A}^v)^\top \mathbf{A}^v = \mathbf{I}, \tag{16}$$

In Eq (16), we define $\mathbf{K} = \sum_{v=1}^m \frac{\epsilon_1}{2}(\mathbf{X}^v - \mathbf{P}^v \mathbf{X}^v - \mathbf{E}^v + \frac{\mathbf{H}_1^v}{\epsilon_1})\mathbf{Z}^v$ and apply singular value decomposition (SVD). From the results of the SVD, we can determine that the optimal solution for $\mathbf{A} = \mathbf{B}\mathbf{D}^\top$, where $\mathbf{B}$ and $\mathbf{D}$ represent the left and right singular vector matrices, respectively.

$\{\mathbf{Q}^v\}$ **Subproblem:** In the scenario where $\mathbf{Q}^v$ is the sole variable in Eq (11), we take the partial derivative of Eq (11) with respect to $\mathbf{Q}^v$ and set it to zero, leading us to express the optimal solution for $\mathbf{Q}^v$ in the following form:

$$\mathbf{Q}^v = (\epsilon_2 \mathbf{Z}^v - \mathbf{H}_2^v)(2\gamma \mathbf{L}_h^v + \epsilon_2 \mathbf{I})^{-1}, \tag{17}$$

$\boldsymbol{\mathcal{S}}$ **Subproblem:** When $\boldsymbol{\mathcal{S}}$ is the only variable, the optimal value of $\boldsymbol{\mathcal{S}}$ can be redefined as a tensor hyperbolic tangent rank optimization problem in the following mathematical form:

$$\min_{\boldsymbol{\mathcal{S}}} \|\boldsymbol{\mathcal{S}}\|_{\mathrm{HTR}} + \frac{\theta}{2} \left\| \boldsymbol{\mathcal{Z}} - \boldsymbol{\mathcal{S}} + \frac{\boldsymbol{\mathcal{Y}}}{\theta} \right\|_F^2 \tag{18}$$

To find the solution for Eq. (18) with respect to $\boldsymbol{\mathcal{S}}$, we first introduce the following theorem related to the tensor optimization problem:

**Theorem 2.** *Let $\boldsymbol{\mathcal{G}} \in \mathbb{R}^{n_1 \times n_2 \times n_3}$ be a tensor, and consider its t-SVD (tensor singular value decomposition) expressed as $\boldsymbol{\mathcal{G}} = \boldsymbol{\mathcal{B}} * \boldsymbol{\mathcal{C}} * \boldsymbol{\mathcal{D}}^\top$. We will analyze the following tensorial hyperbolic tangent rank minimization problem:*

$$\min_{\boldsymbol{\mathcal{S}}} \tau \|\boldsymbol{\mathcal{S}}\|_{\mathrm{HTR}} + \frac{1}{2} \|\boldsymbol{\mathcal{S}} - \boldsymbol{\mathcal{G}}\|_F^2, \tag{19}$$

*The optimal solution for Eq. (19) takes the following mathematical form:*

$$\boldsymbol{\mathcal{S}}^* = \boldsymbol{\mathcal{B}} * ifft(\Theta_{f,\tau}(\boldsymbol{\mathcal{C}}_f^{(k)}), [], 3) * \boldsymbol{\mathcal{D}}^\top, \tag{20}$$

*where $ifft(\Theta_{f,\tau}(\boldsymbol{\mathcal{C}}_f^{(k)}), [], 3)$ is a tensor in which all frontal slices are diagonal matrices, and $\Theta_{f,\tau}(\boldsymbol{\mathcal{C}}_f^{(k)}(ii))$ meet the following condition:*

$$\Theta_{f,\tau}(\boldsymbol{\mathcal{C}}_f^{(k)}(ii)) = \min_{x \geq 0} \frac{1}{2}(x - \boldsymbol{\mathcal{C}}_f^k(ii)^2) + \tau f(x), \tag{21}$$

*where $f(x) = \frac{e^{\delta x} - e^{-\delta x}}{e^{\delta x} + e^{-\delta x}}$.*

---

**Algorithm 1:** Algorithm for solving STONE model

---

**Input:** Multi-view data $\{\mathbf{X}^v\}_{v=1}^m$, trade-off parameters $\alpha, \beta, \gamma$, cluster number $c$ and anchor number $l$.

**Output:** Clustering results

1 Initialize $\{\mathbf{Z}^v, \mathbf{P}^v, \mathbf{Q}^v, \mathbf{E}^v, \mathbf{H}_1^v, \mathbf{H}_2^v\}_{v=1}^m$ with zero matrix, $\boldsymbol{\mathcal{S}} = \boldsymbol{\mathcal{Y}} = 0$, $\epsilon_1 = \epsilon_2 = \theta = 10^{-5}$, $\eta = 2$, $\mu_{max} = \theta_{max} = 10^{10}$, $\mu = 10^{-7}$;

2 **while** *not converge* **do**

3     Compute hyper-Laplacian matrices $\{\mathbf{L}_h^v\}_{v=1}^m$ from $\{\mathbf{A}^v\}_{v=1}^m$;

4     Update $\{\mathbf{Z}^v\}_{v=1}^m$ by solving Eq. (12);

5     Update $\{\mathbf{P}^v\}_{v=1}^m$ by solving Eq. (13);

6     Update $\mathbf{E}$ by solving Eq. (14);

7     Update $\{\mathbf{A}^v\}_{v=1}^m$ by solving Eq. (16);

8     Update $\{\mathbf{Q}^v\}_{v=1}^m$ by solving Eq. (17);

9     Update $\boldsymbol{\mathcal{S}}$ by solving Eq. (20);

10     Update $\boldsymbol{\mathcal{Y}}, \mathbf{H}_1^v, \mathbf{H}_2^v, \epsilon_i$ and $\theta$ by using Eq. (23);

11     Check the convergence conditions: $\|\mathbf{X}^v - \mathbf{A}^v(\mathbf{Z}^v)^\top - \mathbf{P}^v\mathbf{X}^v - \mathbf{E}^v\|_\infty < \mu$ and $\|\boldsymbol{\mathcal{Z}} - \boldsymbol{\mathcal{S}}\|_\infty < \mu$

12 **end**

13 Output clustering results via $k$-means on the left singular vector of the concatenated matrix $\bar{\mathbf{Z}}$.

---

Eq. (21) incorporates a mix of concave and convex functions, which allows for the use of difference of convex programming [67]. This technique facilitates obtaining a closed-form solution.

$$\phi^{iter+1} = \left( \boldsymbol{\mathcal{C}}_f^{(k)}(ii) - \frac{\tau\partial f(\phi^{iter})}{\theta} \right)_+ \tag{22}$$

where $\phi = \Theta_{f, \frac{\beta}{\theta}}(\boldsymbol{\mathcal{C}}_f^{(k)}(ii))$, $f(x) = \frac{e^{\delta x} - e^{-\delta x}}{e^{\delta x} + e^{-\delta x}}$ and $iter$ indicates the iteration count.

**Multipliers and the Penalty Parameters Subproblem:** Finally, $\boldsymbol{\mathcal{Y}}, \mathbf{H}_1^v, \mathbf{H}_2^v, \epsilon_i$ and $\theta$ are updated as follows:

$$\begin{cases} \boldsymbol{\mathcal{Y}} = \boldsymbol{\mathcal{Y}} + \theta(\boldsymbol{\mathcal{Z}} - \boldsymbol{\mathcal{S}}), \\ \mathbf{H}_1^v = \mathbf{H}_1^v + \epsilon_1(\mathbf{X}^v - \mathbf{A}^v(\mathbf{Z}^v)^\top - \mathbf{P}^v\mathbf{X}^v - \mathbf{E}^v), \\ \mathbf{H}_2^v = \mathbf{H}_2^v + \epsilon_2(\mathbf{Z}^v - \mathbf{Q}^v), \\ \mu_i = \min(\eta\epsilon_i, \epsilon_{\max}), i = 1, 2, \\ \theta = \min(\eta\theta, \theta_{\max}). \end{cases} \tag{23}$$

Thus, the solutions for all variables in the STONE model have been optimized. To provide clarity, the complete optimization process is detailed in Algorithm 1.

### A.2 Convergence Proof

The Theorem 1 presented in the main text ensures the convergence of the optimization algorithm. We will now demonstrate the two conditions specified in Theorem 1. To begin, we introduce the following lemma:

**Lemma 1.** *In the context of the real Hilbert space $\mathcal{H}$, we define an inner product $\langle\cdot, \cdot\rangle$ and a norm $|\cdot|$, along with their dual norm $\|\cdot\|^{dual}$. For any element $\mathbf{y}$ within the subdifferential of the function $f(\cdot)$, denoted as $\mathbf{y} \in \partial|\mathbf{x}|$, the subsequent properties are observed: when $\mathbf{x}$ is not the zero vector, the dual norm of $\mathbf{y}$ is exactly 1; when $\mathbf{x}$ is the zero vector, the dual norm of $\mathbf{y}$ does not exceed 1.*

**Lemma 2.** *Consider the function $\mathbf{F}(\mathbf{X}) = f \circ \delta(\mathbf{X})$, where $\delta(\mathbf{X}) = (\sigma_1(\mathbf{X}), \ldots, \sigma_r(\mathbf{X}))$ is the vector of singular values derived from the singular value decomposition (SVD) of $\mathbf{X} \in \mathbb{R}^{m \times n}$, with $r$ being the minimum of $m$ and $n$. The function $f(\cdot) : \mathbb{R}^r \to \mathbb{R}$ is assumed to be differentiable and invariant under permutation of its arguments. The subdifferential of $F(\mathbf{X})$ at the point $\mathbf{X}$ can be expressed as:*

$$\frac{\partial F(\mathbf{X})}{\partial \mathbf{X}} = \mathbf{B}Diag(\partial f(\delta(\mathbf{X})))\mathbf{D}^\top,$$

*where $\partial f(\delta(\mathbf{X})) = \left( \frac{\partial f(\sigma_1(x))}{\partial \mathbf{X}}, \ldots, \frac{\partial f(\sigma_r(x))}{\partial \mathbf{X}} \right)$.*

**Proof of the boundedness of the sequence $\{\mathcal{G}_t\}_{t=1}^{\infty}$:** In the course of the $(t+1)$-th cycle, the mechanism updating $\mathbf{E}_{t+1}^v$ ensures it complies with the necessary first-order optimality criteria. Consequently, it follows that:

$$
\begin{aligned}
0 &\in \beta\partial\|\mathbf{E}_{t+1}^v\|_{2,1} + \epsilon_{1t}\|\mathbf{E}_{t+1}^v - (\mathbf{X}_{t+1}^v - \mathbf{A}^v(\mathbf{Z}_{t+1}^v)^\top - \mathbf{P}_{t+1}^v\mathbf{X}^v + \frac{\mathbf{H}_{1t}^v}{\epsilon_1})\|_F^2 \\
&= \beta\partial\|\mathbf{E}_{t+1}^v\|_{2,1} - \mathbf{H}_{1,t+1}^v,
\end{aligned}
\tag{24}
$$

From Eq. (24), we can derive the following:

$$
\frac{1}{\beta}[\mathbf{H}_{1,t+1}^v]_{:,j} = \partial\|[\mathbf{E}_{t+1}^v]_{:,j}\|_2,
\tag{25}
$$

where $[\mathbf{H}_{1,t+1}^v]_{:,j}$ and $[\mathbf{E}_{t+1}^v]_{:,j}$ correspond to the $j$-th column of the matrices. Furthermore, taking into account the self-duality property of the $\ell_2$ norm and utilizing Lemma 2, we can conclude that $\frac{1}{\beta}[\mathbf{H}_{1,t+1}^v]_{:,j} \le 1$. This further establishes the boundedness of the sequence $[\mathbf{H}_{1,t+1}^v]$. In parallel, the update for $\mathbf{Q}_{t+1}^v$ ensures that $\mathbf{H}_{2,t+1}$ not only meets but also optimizes the first-order optimality criteria. Hence, it can be inferred that the sequence $\mathbf{H}_{2,t+1}^v$ is bounded as well.

Regarding the sequence $\{\mathcal{Y}_{t+1}\}$, the update mechanism for $\mathcal{S}$ guarantees that $\mathcal{S}_{t+1}$ achieves optimality and meets the criteria for first-order optimality. Thus, we have:

$$
\partial\|\mathcal{S}_{t+1}\|_{\mathrm{HTR}} = \mathcal{Y}_{t+1}.
\tag{26}
$$

Furthermore, leveraging the tensor singular value decomposition and Lemma 2, we can derive the following relationship:

$$
\begin{aligned}
\|\partial\|\mathcal{S}_{t+1}\|_{\mathrm{HTR}}\|_F^2 &= \left\|\frac{1}{n}\mathcal{B} * ifft(\partial f(\mathcal{C}_f), [], 3) * \mathcal{D}^T\right\| \\
&= \left\|\frac{1}{n^2}(ifft(\partial f(\mathcal{C}_f), [], 3))\right\|_F^2 = \left\|\frac{1}{n^3}(\partial f(\mathcal{C}_f))\right\|_F^2 \\
&\le \frac{1}{n^3}\sum_{k=1}^{n}\sum_{j=1}^{min(n,m)}[(\partial f(\mathcal{C}_f^k(jj)))]^2.
\end{aligned}
\tag{27}
$$

This observation indicates that $\partial\|\mathcal{S}_{f,t+1}\|_{\mathrm{HTR}}$ has an upper bound, which in turn allows us to deduce that the sequenc $\{\mathcal{Y}_{t+1}\}$ is bounded.

Based on the iterative procedures described in Algorithm 1, we can derive the following inequality relationships:

$$
\begin{aligned}
&\mathcal{L}(\mathbf{Z}_{t+1}^v, \mathbf{E}_{t+1}^v, \mathbf{P}_{t+1}^v, \mathbf{A}_{t+1}^v, \mathbf{Q}_{t+1}^v, \mathcal{S}_{t+1}, \mathbf{H}_{1,t}^v, \mathbf{H}_{2,t}^v, \mathcal{Y}_t, \theta_t, \epsilon_{1,t}, \epsilon_{2,t}) \\
&\le \mathcal{L}(\mathbf{Z}_t^v, \mathbf{E}_t^v, \mathbf{P}_t^v, \mathbf{A}_t^v, \mathbf{Q}_t^v, \mathcal{S}_t, \mathbf{H}_{1,t}^v, \mathbf{H}_{2,t}^v, \mathcal{Y}_t, \theta_t, \epsilon_{1,t}, \epsilon_{2,t}) \\
&= \mathcal{L}(\mathbf{Z}_t^v, \mathbf{E}_t^v, \mathbf{P}_t^v, \mathbf{A}_t^v, \mathbf{Q}_t^v, \mathcal{S}_t, \mathbf{H}_{1,t-1}^v, \mathbf{H}_{2,t-1}^v, \mathcal{Y}_{t-1}, \theta_{t-1}, \epsilon_{1,t-1}, \epsilon_{2,t-1}) \\
&+ \frac{\theta_t - \theta_{t-1}}{2\theta_{t-1}^2}\|\mathcal{Y}_t - \mathcal{Y}_{t-1}\|_F^2 + \frac{\epsilon_{1,t} - \epsilon_{1,t-1}}{2\epsilon_{1,t-1}^2}\|\mathbf{H}_{1,t}^v - \mathbf{H}_{1,t-1}^v\|_F^2 \\
&+ \frac{\epsilon_{2,t} - \epsilon_{2,t-1}}{2\epsilon_{2,t-1}^2}\|\mathbf{H}_{2,t}^v - \mathbf{H}_{2,t-1}^v\|_F^2
\end{aligned}
\tag{28}
$$

Consequently, by adding up both sides of Eq. (28) over the range from $t=1$ to $t=n$, we deduce the following consequence:

$$
\begin{aligned}
&\mathcal{L}(\mathbf{Z}_{t+1}^v, \mathbf{E}_{t+1}^v, \mathbf{P}_{t+1}^v, \mathbf{A}_{t+1}^v, \mathbf{Q}_{t+1}^v, \mathcal{S}_{t+1}, \mathbf{H}_{1,t}^v, \mathbf{H}_{2,t}^v, \mathcal{Y}_t, \theta_t, \epsilon_{1,t}, \epsilon_{2,t}) \\
&\le \mathcal{L}(\mathbf{Z}_1^v, \mathbf{E}_1^v, \mathbf{P}_1^v, \mathbf{A}_1^v, \mathbf{Q}_1^v, \mathcal{S}_1, \mathbf{H}_{1,0}^v, \mathbf{H}_{2,0}^v, \mathcal{Y}_0, \theta_0, \epsilon_{1,0}, \epsilon_{2,0}) \\
&+ \sum_{t=1}^{n}\frac{\theta_t - \theta_{t-1}}{2\theta_{t-1}^2}\|\mathcal{Y}_t - \mathcal{Y}_{t-1}\|_F^2 \\
&+ \sum_{t=1}^{n}\frac{\epsilon_{1,t} - \epsilon_{1,t-1}}{2\epsilon_{1,t-1}^2}\|\mathbf{H}_{1,t}^v - \mathbf{H}_{1,t-1}^v\|_F^2 \\
&+ \sum_{t=1}^{n}\frac{\epsilon_{2,t} - \epsilon_{2,t-1}}{2\epsilon_{2,t-1}^2}\|\mathbf{H}_{2,t}^v - \mathbf{H}_{2,t-1}^v\|_F^2
\end{aligned}
\tag{29}
$$

Given the finite nature of the initial $\mathcal{L}(\mathbf{Z}_1^v, \mathbf{E}_1^v, \mathbf{P}_1^v, \mathbf{A}_1^v, \mathbf{Q}_1^v, \boldsymbol{\mathcal{S}}_1, \mathbf{H}_{1,0}^v, \mathbf{H}_{2,0}^v, \boldsymbol{\mathcal{Y}}_0, \theta_0, \epsilon_{1,0}, \epsilon_{2,0})$ evaluated at the starting points and the boundedness of the sequences $\{\boldsymbol{\mathcal{Y}}_t\}$, $\{\mathbf{H}_{1,t}\}$, $\{\mathbf{H}_{2,t}\}$, along with the boundedness of the incremental sums involving $\sum_{t=1}^n \frac{\theta_t - \theta_{t-1}}{2\theta_{t-1}^2}$, $\sum_{t=1}^n \frac{\epsilon_{1,t} - \epsilon_{1,t-1}}{2\epsilon_{1,t-1}^2}$ and $\sum_{t=1}^n \frac{\epsilon_{2,t} - \epsilon_{2,t-1}}{2\epsilon_{2,t-1}^2}$, we deduce that sequence $\mathcal{L}$ remains bounded at iteration $t+1$. Additionally, since the norm $\|\boldsymbol{\mathcal{S}}_{t+1}\|_{\mathrm{HTR}}$ is bounded, it follows that the singular values of $\boldsymbol{\mathcal{S}}_{t+1}$ are also constrained. Continuing with the equation:

$$
\begin{aligned}
\|\boldsymbol{\mathcal{S}}_{t+1}\|_F^2 &= \frac{1}{n_3} \|\boldsymbol{\mathcal{S}}_{t+1}\|_F^2 \\
&= \frac{1}{n_3} \sum_{i=1}^{n_3} \sum_{j=1}^{\min(n_1,n_2)} [((\boldsymbol{\mathcal{C}}_f^{(i)}(jj))]^2,
\end{aligned}
\tag{30}
$$

It is verified that the sequence $\{\boldsymbol{\mathcal{S}}_{t+1}\}$ has finite limits. Moreover, it is evident that the sequences $\{\mathbf{A}_{t+1}^v\}$, $\{\mathbf{Z}_{t+1}^v\}$, $\{\mathbf{Q}_{t+1}^v\}$, and $\{\mathbf{P}_{t+1}^v\}$ are also finite.

Continuing from the earlier findings, we can assert that the sequence $\mathcal{G}_t$, as yielded by Algorithm 1, is bounded for every component within it.

**Proof of convergence of accumulation points to stationary KKT points:** As per the Weierstrass-Bolzano theorem, it is guaranteed that the sequence $\{\mathcal{G}_t\}_{t=1}^\infty$ contains at least one accumulation point, which we label as $\mathcal{G}_t^* = \{\mathbf{Z}_t^v, \mathbf{E}_t^v, \mathbf{P}_t^v, \mathbf{A}_t^v, \boldsymbol{\mathcal{Y}}_t, \mathbf{H}_{1t}^v, \mathbf{H}_{2t}^v, \boldsymbol{\mathcal{S}}_t\}_{t=1}^\infty$. From this, we can infer that:

$$
\begin{aligned}
&\lim_{t\to\infty} (\mathbf{Z}_t^v, \mathbf{E}_t^v, \mathbf{P}_t^v, \mathbf{A}_t^v, \mathbf{Q}_t^v, \boldsymbol{\mathcal{S}}_t, \mathbf{H}_{1t}^v, \mathbf{H}_{2t}^v, \boldsymbol{\mathcal{Y}}_t) \\
&= (\mathbf{Z}_*^v, \mathbf{E}_*^v, \mathbf{P}_*^v, \mathbf{A}_*^v, \mathbf{Q}_v^*, \boldsymbol{\mathcal{S}}_*, \mathbf{H}_{1*}^v, \mathbf{H}_{2*}^v, \boldsymbol{\mathcal{Y}}_*).
\end{aligned}
\tag{31}
$$

Adhering to the update mechanism for $\boldsymbol{\mathcal{Y}}$, we arrive at the subsequent expression:

$$
\boldsymbol{\mathcal{Z}}_{t+1} - \boldsymbol{\mathcal{S}}_{t+1} = (\boldsymbol{\mathcal{Y}}_{t+1} - \boldsymbol{\mathcal{Y}}_t)/\theta_t,
\tag{32}
$$

Since $\theta_t$ approach infinity as $t$ goes to infinity, and considering that the sequence $\{\boldsymbol{\mathcal{Y}}_t\}$ is bounded, we can use the properties of limits to derive:

$$
\lim_{t\to\infty} \boldsymbol{\mathcal{Z}}_{t+1} - \boldsymbol{\mathcal{S}}_{t+1} = \lim_{t\to\infty} (\boldsymbol{\mathcal{Y}}_{t+1} - \boldsymbol{\mathcal{Y}}_t)/\theta_t = 0
\tag{33}
$$

By applying the analogous update processes for $\mathbf{H}_1$ and $\mathbf{H}_2$, we can formulate the subsequent relationship:

$$
\begin{cases}
\mathbf{X}_{t+1}^v - \mathbf{A}_{t+1}^v (\mathbf{Z}_{t+1}^v)^\top - \mathbf{P}_{t+1}^v \mathbf{X}^v - \mathbf{E}_{t+1}^v = \dfrac{\mathbf{H}_{1,t+1}^v - \mathbf{H}_{1,t}^v}{\epsilon_{1,t}}, \\[2mm]
\mathbf{Z}_{t+1}^v - \mathbf{Q}_{t+1}^v = \dfrac{\mathbf{H}_{2,t+1}^v - \mathbf{H}_{2,t}^v}{\epsilon_{2,t}}.
\end{cases}
\tag{34}
$$

In the same vein, considering that the sequences $\{\mathbf{H}_{1,t}^v\}$ and $\{\mathbf{H}_{2,t}^v\}$ remain bounded, while $\epsilon_{1,t}$ and $\epsilon_{2,t}$ increase indefinitely as $t$ approaches infinity, we can infer the following conclusions based on limit properties:

$$
\begin{cases}
\lim_{t\to\infty} \mathbf{X}_{t+1}^v - \mathbf{A}_{t+1}^v (\mathbf{Z}_{t+1}^v)^\top - \mathbf{P}_{t+1}^v \mathbf{X}^v - \mathbf{E}_{t+1}^v = \lim_{t\to\infty} \dfrac{\mathbf{H}_{1,t+1}^v - \mathbf{H}_{1,t}^v}{\epsilon_{1,t}} = 0, \\[2mm]
\lim_{t\to\infty} \mathbf{Z}_{t+1}^v - \mathbf{Q}_{t+1}^v = \lim_{t\to\infty} \dfrac{\mathbf{H}_{2,t+1}^v - \mathbf{H}_{2,t}^v}{\epsilon_{2,t}} = 0.
\end{cases}
\tag{35}
$$

Based on Eqs. (33) and (35), we can deduce that in the limit, $\boldsymbol{\mathcal{Z}}_*$ is equal to $\boldsymbol{\mathcal{Z}}_*$, while $\mathbf{X}^v$ and $\mathbf{E}^v$ exhibit a specific linear relationship, and $\mathbf{Z}_*^v$ equals $\mathbf{Q}_*^v$. Additionally, since $\boldsymbol{\mathcal{S}}_{t+1}$, $\mathbf{E}_{t+1}^v$ and $\mathbf{Q}_{t+1}^v$ satisfy the first-order optimality conditions, we can conclude that:

$$
\begin{cases}
0 \in \partial \|\boldsymbol{\mathcal{S}}_{t+1}\|_{\mathrm{HTR}} - \boldsymbol{\mathcal{Y}}_{t+1} \Rightarrow \boldsymbol{\mathcal{Y}}_* = \partial \|\boldsymbol{\mathcal{S}}_*\|_{\mathrm{HTR}} \\
0 \in \beta \partial \|\mathbf{E}_{t+1}^v\|_{2,1} - \mathbf{H}_{1,t+1}^v \Rightarrow \mathbf{H}_{1,*}^v = \beta \partial \|\mathbf{E}_*^v\|_{2,1} \\
0 \in \gamma \partial \mathrm{Tr}(\mathbf{Q}_{t+1}^v \mathbf{L}_{t+1}^v (\mathbf{Q}_{t+1}^v)^\top) - \mathbf{H}_{2,t+1}^v \Rightarrow \mathbf{H}_{2,*}^v = \gamma \partial \mathrm{Tr}(\mathbf{Q}_*^v \mathbf{L}_*^v (\mathbf{Q}_*^v)^\top)
\end{cases}
\tag{36}
$$

Consequently, the accumulation points of the sequence $\mathcal{G}_t$ produced by Algorithm 1 fulfill the KKT conditions.

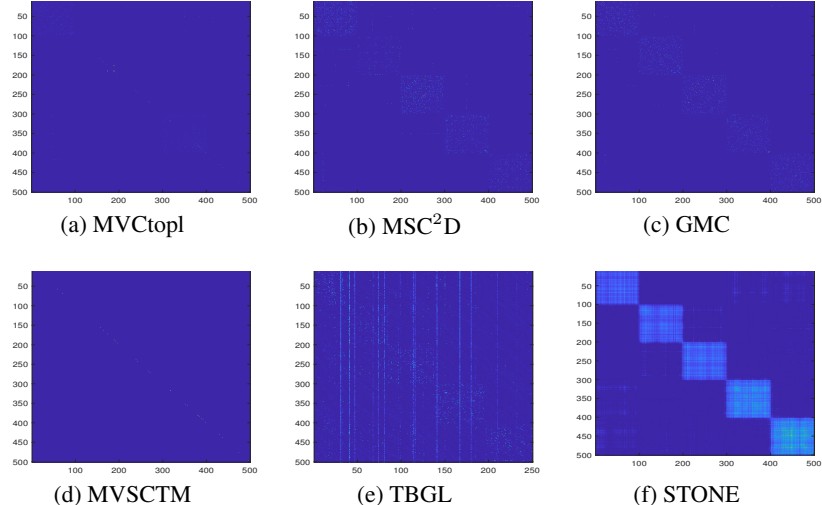

Figure 7: Contrasting Consensus Affinity Matrices: STONE vs. SOTA on NGs Dataset.

### A.3 Additional Experimental Results

This section outlines additional experimental results, including visualizations of block diagonal structures, t-SNE, and some extra experiments mentioned in the text, such as parameter sensitivity, convergence curves, and ablation studies.

**Comparison of Block Diagonal Structures in Affinity Matrices:** In this section, we examine the block diagonal structures of affinity matrices learned by the STONE model alongside several other state-of-the-art multi-view clustering methods. The comparative results on the NGs dataset are illustrated in Figure 7. Notably, the consensus affinity matrix generated by the STONE model clearly displays a well-defined block diagonal structure, while those produced by other approaches frequently exhibit many spurious connections. This reinforces the effectiveness of the STONE model in leveraging rich information from multi-view data through the synergistic integration of discriminative anchor point learning, local structural information extraction, and the utilization of tensor data priors, ultimately enhancing clustering performance.

**Analysis of Multi-View Advantages Over Single-View:** To demonstrate the STONE model's capability in leveraging the rich information inherent in multi-view data, Figure 8 displays the t-SNE visualizations for each individual view alongside the integrated consensus graph. Notably, the consensus graph presents a more distinct clustering pattern when compared to the individual view graphs, which aids in the more precise segregation of the MSRCV1 dataset into seven unique classes. This finding highlights the effectiveness of the STONE method in synthesizing multi-view data for improved clustering performance.

**Experimental Results for More Datasets:** Due to space constraints, the main text presents only partial results for some experiments. Here, we provide the complete results for all datasets, including parameter sensitivity analyses, convergence curves, and ablation studies. Specifically, Figure 9, Figure 10 and Figure 11 showcase the sensitivity of the STONE model to the parameters $\delta$, the number of anchors $l$, and the balancing parameters. Figure 12 illustrates the convergence curves for eight datasets, while Table 6-Table 9 summarize the ablation studies for the three loss terms $\mathcal{L}_{EAD}$, $\mathcal{L}_{RE}$ and $\mathcal{L}_{AHR}$ of STONE across all datasets.

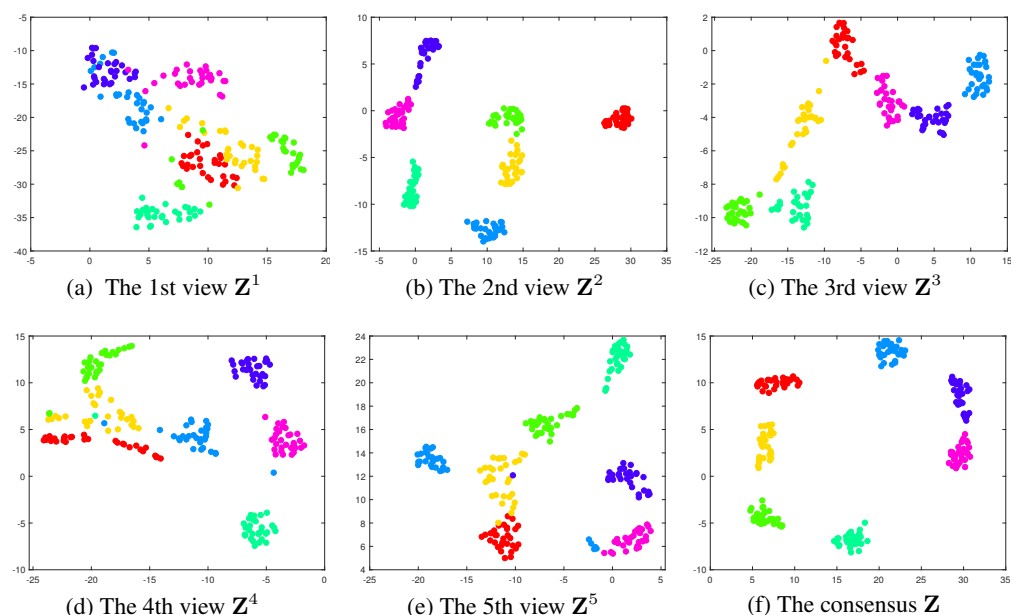

| (a) The 1st view $\mathbf{Z}^1$ | (b) The 2nd view $\mathbf{Z}^2$ | (c) The 3rd view $\mathbf{Z}^3$ |
| (d) The 4th view $\mathbf{Z}^4$ | (e) The 5th view $\mathbf{Z}^5$ | (f) The consensus $\mathbf{Z}$ |

Figure 8: Comparative t-SNE Visualization Analysis: View-Specific Graphs vs. Consensus Graph.

Table 6: Analysis of STONE Model Ablation on NGs and BBCSport Datasets.

| Datasets | | | NGs | | | | | BBCSport | | | | |
|---|---|---|---|---|---|---|---|---|---|---|---|---|
| $\mathcal{L}_{EAD}$ | $\mathcal{L}_{RE}$ | $\mathcal{L}_{AHR}$ | ACC | NMI | PUR | F-score | ARI | ACC | NMI | PUR | F-score | ARI |
| ✔ | | | 0.438 | 0.291 | 0.438 | 0.348 | 0.183 | 0.800 | 0.690 | 0.800 | 0.691 | 0.600 |
| | ✔ | | 0.208 | 0.012 | 0.212 | 0.329 | 0.000 | 0.831 | 0.712 | 0.831 | 0.711 | 0.624 |
| | | ✔ | 0.596 | 0.473 | 0.636 | 0.532 | 0.396 | 0.996 | 0.987 | 0.996 | 0.996 | 0.995 |
| ✔ | ✔ | | 0.960 | 0.897 | 0.960 | 0.924 | 0.905 | 0.818 | 0.715 | 0.818 | 0.691 | 0.598 |
| ✔ | | ✔ | 0.534 | 0.397 | 0.572 | 0.449 | 0.274 | 0.368 | 0.137 | 0.474 | 0.296 | 0.084 |
| | ✔ | ✔ | 0.458 | 0.311 | 0.502 | 0.404 | 0.196 | 0.358 | 0.134 | 0.474 | 0.292 | 0.083 |
| ✔ | ✔ | ✔ | **1.000** | **1.000** | **1.000** | **1.000** | **1.000** | **1.000** | **1.000** | **1.000** | **1.000** | **1.000** |

Table 7: Analysis of STONE Model Ablation on HW and Scene15 Datasets.

| Datasets | | | HW | | | | | Scene15 | | | | |
|---|---|---|---|---|---|---|---|---|---|---|---|---|
| $\mathcal{L}_{EAD}$ | $\mathcal{L}_{RE}$ | $\mathcal{L}_{AHR}$ | ACC | NMI | PUR | F-score | ARI | ACC | NMI | PUR | F-score | ARI |
| ✔ | | | 0.988 | 0.974 | 0.988 | 0.976 | 0.974 | 0.589 | 0.557 | 0.597 | 0.480 | 0.440 |
| | ✔ | | 0.977 | 0.951 | 0.977 | 0.955 | 0.950 | 0.787 | 0.844 | 0.802 | 0.714 | 0.693 |
| | | ✔ | 0.988 | 0.974 | 0.988 | 0.976 | 0.974 | 0.328 | 0.297 | 0.359 | 0.228 | 0.168 |
| ✔ | ✔ | | 0.881 | 0.841 | 0.881 | 0.815 | 0.794 | 0.724 | 0.802 | 0.760 | 0.655 | 0.629 |
| ✔ | | ✔ | 0.983 | 0.971 | 0.983 | 0.968 | 0.965 | 0.713 | 0.800 | 0.764 | 0.656 | 0.630 |
| | ✔ | ✔ | 0.854 | 0.841 | 0.854 | 0.786 | 0.762 | 0.678 | 0.807 | 0.746 | 0.687 | 0.663 |
| ✔ | ✔ | ✔ | **1.000** | **1.000** | **1.000** | **1.000** | **1.000** | **0.977** | **0.962** | **0.977** | **0.958** | **0.955** |

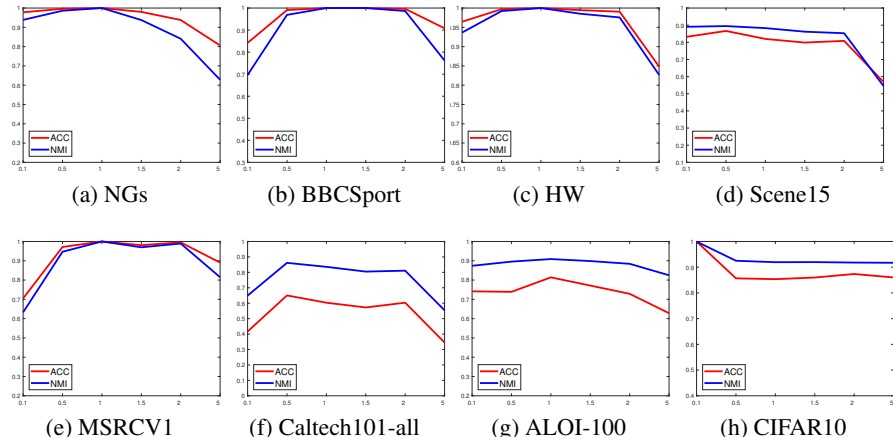

Figure 9: Impact of Parameter $\delta$ on the STONE Model on Eight Datasets.

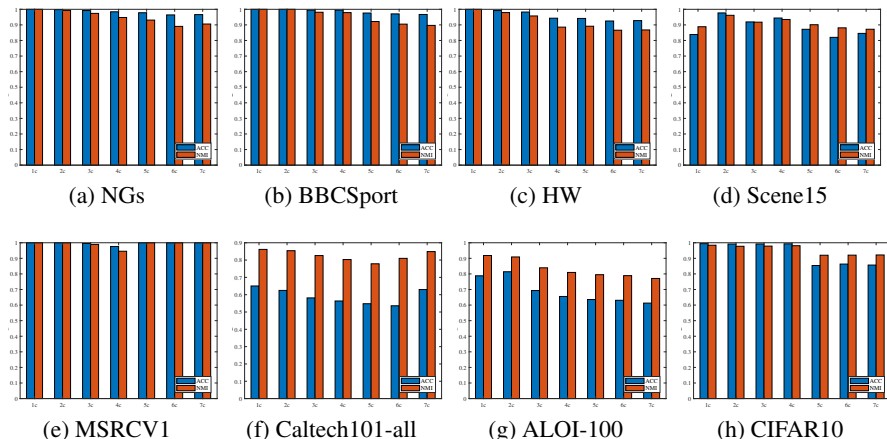

Figure 10: The Influence of Anchor Quantity on STONE Model Performance Across Eight Datasets.

Table 8: Analysis of STONE Model Ablation on MSRCV1 and ALOI-100 Datasets.

| Datasets | | | MSRCV1 | | | | | ALOI-100 | | | | |
| --- | --- | --- | --- | --- | --- | --- | --- | --- | --- | --- | --- | --- |
| $\mathcal{L}_{EAD}$ | $\mathcal{L}_{RE}$ | $\mathcal{L}_{AHR}$ | ACC | NMI | PUR | F-score | ARI | ACC | NMI | PUR | F-score | ARI |
| ✔ | | | 0.148 | 0.030 | 0.171 | 0.243 | 0.001 | 0.611 | 0.759 | 0.630 | 0.470 | 0.464 |
| | ✔ | | 0.205 | 0.033 | 0.214 | 0.175 | -0.002 | 0.592 | 0.757 | 0.628 | 0.444 | 0.438 |
| | | ✔ | 0.148 | 0.030 | 0.171 | 0.243 | 0.001 | 0.488 | 0.669 | 0.530 | 0.234 | 0.222 |
| ✔ | ✔ | | 0.976 | 0.946 | 0.976 | 0.952 | 0.944 | 0.545 | 0.778 | 0.614 | 0.440 | 0.433 |
| ✔ | | ✔ | 0.786 | 0.631 | 0.786 | 0.630 | 0.570 | 0.580 | 0.776 | 0.625 | 0.447 | 0.439 |
| | ✔ | ✔ | 0.571 | 0.384 | 0.571 | 0.407 | 0.306 | 0.557 | 0.737 | 0.592 | 0.391 | 0.383 |
| ✔ | ✔ | ✔ | **1.000** | **1.000** | **1.000** | **1.000** | **1.000** | **0.814** | **0.909** | **0.849** | **0.736** | **0.733** |

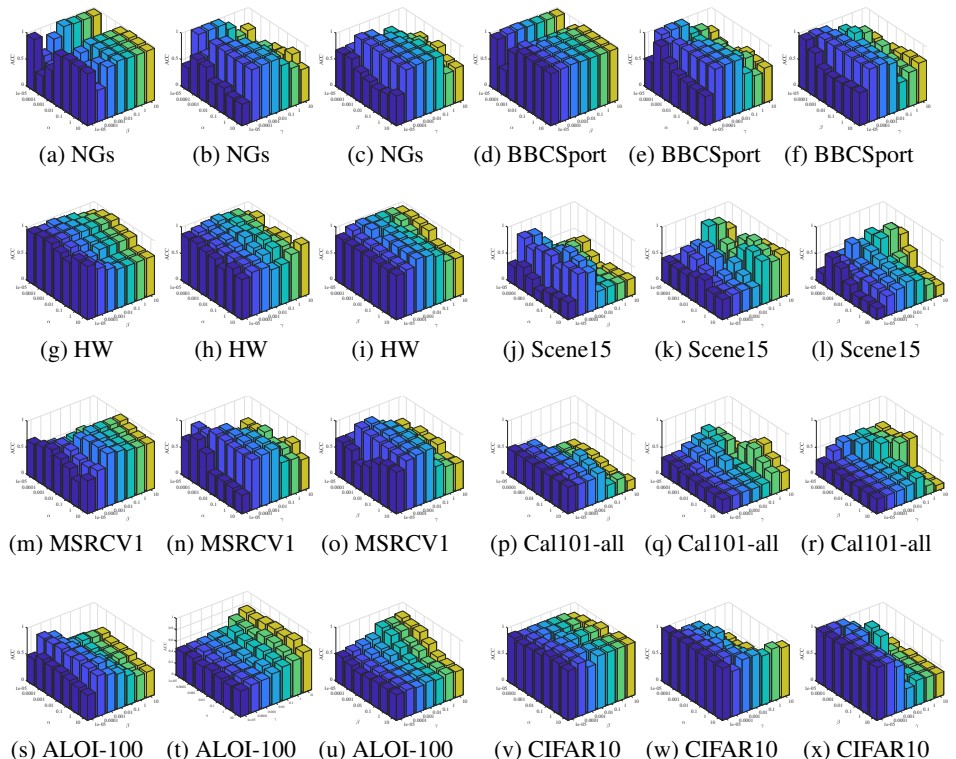

Figure 11: Sensitivity Analysis of the STONE Model to Parameters $\alpha$, $\beta$ and $\gamma$ on Eight Datasets.

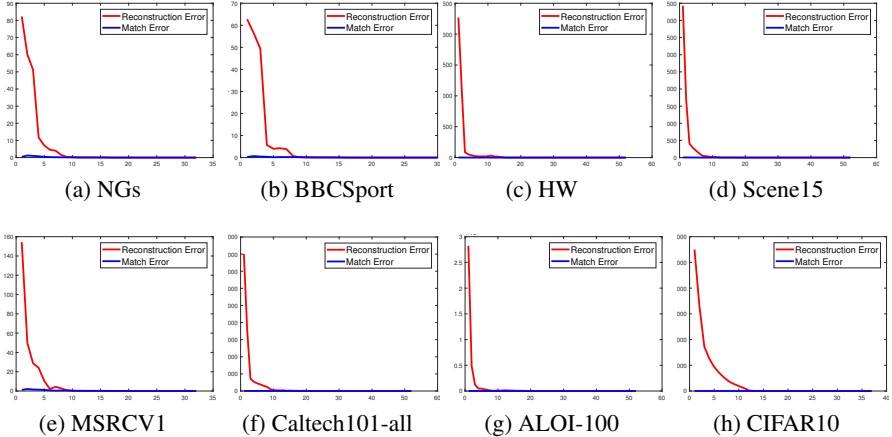

Figure 12: Convergence Curves of STONE Model on Eight Datasets.

Table 9: Analysis of STONE Model Ablation on Caltech101-all and CIFAR10 Datasets.

| Datasets | | | Caltech101-all | | | | | CIFAR10 | | | | |
|---|---|---|---|---|---|---|---|---|---|---|---|---|
| $\mathcal{L}_{EAD}$ | $\mathcal{L}_{RE}$ | $\mathcal{L}_{AHR}$ | ACC | NMI | PUR | F-score | ARI | ACC | NMI | PUR | F-score | ARI |
| ✔ | | | 0.185 | 0.368 | 0.365 | 0.182 | 0.168 | 0.833 | 0.782 | 0.833 | 0.733 | 0.703 |
| | ✔ | | 0.475 | 0.786 | 0.717 | 0.345 | 0.334 | 0.931 | 0.876 | 0.931 | 0.874 | 0.860 |
| | | ✔ | 0.271 | 0.477 | 0.467 | 0.206 | 0.191 | 0.994 | 0.983 | 0.994 | 0.988 | 0.987 |
| ✔ | ✔ | | 0.499 | 0.799 | 0.753 | 0.378 | 0.368 | 0.830 | 0.867 | 0.857 | 0.829 | 0.809 |
| ✔ | | ✔ | 0.297 | 0.531 | 0.527 | 0.244 | 0.229 | 0.830 | 0.867 | 0.857 | 0.829 | 0.809 |
| | ✔ | ✔ | 0.516 | 0.740 | 0.732 | 0.405 | 0.394 | 0.884 | 0.821 | 0.884 | 0.805 | 0.783 |
| ✔ | ✔ | ✔ | **0.609** | **0.834** | **0.815** | **0.494** | **0.485** | **0.994** | **0.984** | **0.994** | **0.989** | **0.988** |

