# OpenReview forum: "From Dictionary to Tensor: A Scalable Multi-View Subspace Clustering Framework with Triple Information Enhancement"
_NeurIPS.cc/2024/Conference — NeurIPS 2024 poster_

### Official Review · Reviewer_eQ9V · 2024-07-03

**Soundness:** 2
**Presentation:** 3
**Contribution:** 1
**Rating:** 3
**Confidence:** 5

**Summary:**

This paper proposes a scalable tensor-based multi-view subspace clustering model by using triple information enhancement, which aims to reduce the computational complexity and the bias from the real rank minimization.

**Strengths:**

（1）This paper has provided significant proof of the algorithm's convergence.
（2）Comprehensive experiments have been conducted for the investigation of effectiveness and performance.

**Weaknesses:**

1) In LatLRR, P is constraint by nuclear norm. In model (5), the authors indicate that P is constraint by weighted Frobenius norm. The authors can explain the reasons.
2) On line 158, “Z_f^k denotes the k-th frontal slice of Z” should be changed into “Z_f^k denotes the k-th frontal slice of Z_f”.
3) On line 159, “Z=U_f V_f W_f” should be changed into “Z_f=U_f V_f W_f”. The authors could also distinguish the matrix multiplication and tensor multiplication.
4) In Eq. (8), “Tr(Z^v L_h^v Z^v)” should be changed into “Tr(〖〖(Z〗^v)〗^T L_h^v Z^v)”.
5) In Eq. (8), is the hyper-Laplacian matrix L_h^v constructed by anchor subspace Z^v or feature X^v? If L_h^v is constructed by Z^v, the authors could introduce a consistent indicator matrix instead of regularize the Z^v in the trace norm.
6) In Remark 2, as x→0, these small singular values may be caused by the noise. However, the proposed HTR would amplify these small singular values by f(x), resulting in less robustness.
7) In Eq. (10), “α/2” should be changed into “α”.
8) In Eq. (10), “Tr(Q^v L_h^v 〖〖(Q〗^v)〗^T)” should be changed into “Tr(〖〖(Q〗^v)〗^T L_h^v Q^v)”.
9) On line 233, “d” should be changed into “d_v”.
10) How does the HTR effect the proposed model? In Ablation Study, the authors could provide more experiments for verifying the effectiveness of the HTR.

**Questions:**

1) In Eq. (26), how is the parameter “β derived?

**Limitations:**

Please refer to the above "Weaknesses".

---

> ### Author Rebuttal · Authors · 2024-08-06
>
> We appreciate your careful review and insightful feedback on our manuscript. We have thoroughly considered each of your comments and have addressed them in detail below:
>
> **Weakness 1:** Why does LatLRR use the nuclear norm for P, but the proposed method uses the weighted Frobenius norm?
>
> **A1:** We replaced the nuclear norm used in LatLRR with the weighted Frobenius norm in our model because the nuclear norm in LatLRR is intended for feature extraction tasks. Since our manuscript focuses solely on clustering and does not involve feature extraction, applying the nuclear norm to P^v would increase computational complexity without significant benefits. Therefore, we opted to relax the constraint on P^v to the Frobenius norm, which has been shown in previous literature [41-43] to effectively preserve the block-diagonal low-rank structure. Additionally, considering the linear combination of different views, we applied a weighted Frobenius norm to the matrices  {P^v}, with the weights for different views set to 1. This approach is briefly introduced in the final paragraph of Section 2 of our manuscript, and we will provide a more detailed discussion of this choice in the revised manuscript.
>
> **Weakness 2:** On line 158, change $\mathcal{Z}$ to $\mathcal{Z}_f^k$.
>
> **A2:** We will correct this typo in the revised manuscript.
>
> **Weakness 3:**  On line 159, $\mathcal{Z}$ should be changed into $\mathcal{Z}_f^k$. The authors could also distinguish the matrix multiplication and tensor multiplication.
>
> **A3:** We will correct this typo and clarify the difference between matrix and tensor multiplication in the revised manuscript. Matrix multiplication involves dot products of rows and columns in 2D matrices, while tensor multiplication handles multi-dimensional arrays.
>
> **Weakness 4:** In Eq. (8), $Tr(\mathbf Z^v \mathbf L_h^v \mathbf Z^v)$ should be changed into $Tr({(\mathbf Z^v)}^T \mathbf L_h^v \mathbf Z^v)$.
>
> **A4：** We will correct the typo in Eq. (8) to align with the standard format for Laplacian manifold regularization.
>
> **Weakness 5:**  In Eq. (8), is the hyper-Laplacian matrix $\mathbf{L}_h^v$ constructed from $\mathbf{Z}^v$ or $\mathbf{X}^v$? If from $\mathbf{Z}^v$, why not use a consistent indicator matrix instead of regularizing $\mathbf{Z}^v$ in the trace norm?
>
> **A5:** Thank you for your comment. We would like to clarify that in Eq. (8), the hyper-Laplacian matrix $\mathbf{L}_h^v$ is constructed based on the anchor hypergraph $\mathbf{S}^v$, which in turn is derived from the anchor representations $\mathbf{Z}^v$, rather than directly from the anchor subspace $\mathbf{Z}^v$ or the feature matrix $\mathbf{X}^v$. The hyperanchor graph Laplacian manifold regularization builds upon traditional Laplacian regularization [47][48] and is designed to capture high-order local manifold information between anchor representations, which is crucial for the regularization of the subspace representation $\mathbf{Z}^v$. We will provide further clarification on this in the revised manuscript.
>
> **Weakness 6:** In Remark 2, as x→0, these small singular values may be caused by the noise. However, the proposed HTR would amplify these small singular values by f(x), resulting in less robustness.
>
> **A6.** Thank you for your comment. It is true that small singular values may be caused by noise as x→0. However, Remark 2 should clarify that our proposed HTR method does not amplify these small singular values. Instead, the HTR method applies a stronger penalty to these values, which helps to mitigate their impact and improve robustness. The HTR method penalizes small singular values more heavily, thereby reducing their influence rather than amplifying it. This is discussed in detail in the REMARK 2 of the manuscript (including the illustration in Figure 2). We will provide additional clarification on this in the revised manuscript.
>
> **Weakness 7:** In Eq. (10), $\frac{\alpha}{2}$ should be changed into $\alpha$.
>
> **A7:** We will correct this typo in the revised manuscript.
>
> **Weakness 8:**  In Eq. (10), $Tr(\mathbf{Q}^v \mathbf{L}_h^v (\mathbf{Q}^v)^T)$ should be changed into $Tr(  (\mathbf{Q}^v)^T)\mathbf{L}_h^v\mathbf{Q}^v)$.
>
> **A8:** Thank you for bringing this to our attention. We would like to respectfully note that the expression in Eq. (10) follows the standard convention for Laplacian manifold regularization. This format is commonly used in the literature[47][48], and we believe it aligns with established practices.
>
> **Weakness 9:** On line 233, “d” should be changed into “d_v”.
>
> **A9:** We will correct this typo in the revised manuscript.
>
> **Weakness 10:** How does the HTR affect the proposed model? In the Ablation Study, the authors could provide more experiments to verify the effectiveness of the HTR.
>
> **A10:** HTR is a novel tensor low-rank constraint in our STONE model, designed to capture high-order correlations and manage variations in tensor singular values. To assess its impact, we conducted ablation studies comparing the full STONE model with a version excluding HTR (STONE-v1). The results are summarized in the table below:
>
> | **Datasets**      | NGs           | BBCSport       | HW            | Scene15       | MSRCV1        | ALOI-100      | Cal101-all | CIFAR10       |
> |:---------------:|:--------------:|:--------------:|:-------------:|:-------------:|:-------------:|:-------------:|:--------------:|:-------------:|
> | STONE-v1 | 0.379±0.000 | 0.648±0.000 | 0.740±0.000 | 0.629±0.000 | 0.469±0.000 | 0.600±0.000 | 0.319±0.000 | 0.501±0.000 |
> | **Ours**        | **1.000±0.000** | **1.000±0.000** | **1.000±0.000** | **0.977±0.000** | **1.000±0.000** | **0.814±0.000** | **0.650±0.000** | **0.994±0.000** |
>
> **Question 1:** In Eq. (26), how is the parameter $\beta$ derived?
>
> **A11:** $\beta$ in Eq. (26) is a typo and should be $\tau$ as described in Eqs. (20)–(25). $\tau$ is a non-negative real number, as discussed in Lemma 1. We will correct this in the revised manuscript.

---

> > ### Author Response · Authors · 2024-08-14
> >
> > Dear Reviewer eQ9V,
> >
> > We sincerely appreciate your constructive feedback. We have addressed all your concerns regarding our manuscript. As the rebuttal deadline approaches, we would like to kindly ask if you have any further questions or require additional clarification. If not, we would be very grateful if you could reconsider your score.
> >
> > Thank you for your time and consideration.
> >
> > Best regards,
> >
> > Author

---

### Official Review · Reviewer_iXTf · 2024-07-08

**Soundness:** 4
**Presentation:** 4
**Contribution:** 3
**Rating:** 7
**Confidence:** 5

**Summary:**

The manuscript introduces a novel tensor-based multi-view clustering algorithm designed to address three critical limitations of existing approaches: high computational complexity arising from reliance on complete dictionaries, inaccurate subspace representation due to disregarding local geometric information, and inadequate penalization of singular values associated with noise. To overcome these challenges, the authors propose the STONE framework, which integrates enhanced anchor dictionary learning, anchor hypergraph Laplacian regularization, and an improved hyperbolic tangent function for more accurate tensor rank approximation. Experimental results demonstrate that STONE surpasses current state-of-the-art methods in both effectiveness and efficiency.

**Strengths:**

1. The proposed method explores rich information in multi-view data from the perspectives of dictionary representation, subspace representation, and tensor representation to enhance clustering performance, offering an interesting perspective for research in the field of multi-view clustering.

2. The proposed method achieves optimal clustering performance while maintaining high computational efficiency, ensuring its scalability.

3. The authors conducted clustering experiments on 8 multi-view datasets covering various types and scales, yielding compelling results.

4. The experimental and theoretical analyses are comprehensive, providing both empirical and theoretical support for the effectiveness of the proposed method.

**Weaknesses:**

1. The introduction of Enhanced Anchor Dictionary (EAD) lacks clarity. Could the authors clarify the fundamental differences between EAD, traditional dictionary learning, and anchor dictionary learning? Additionally, how does the methodology incorporate hidden data?

2. In Section 4.3, the authors reference Figure 11 in the first part, but the figure numbering is incorrect and needs to be corrected.

3. From Table 2 in the manuscript, it appears that most methods have a standard deviation of 0.000. Could the authors explain why these methods exhibit no standard deviation?

4. Based on Figure 10, why does the performance of the proposed method not exhibit a linear positive correlation with the number of anchor points? Typically, increasing the number of anchor points should lead to acquiring more valuable information.

5. The pseudocode displayed for Algorithm 1 in the appendix has formula indices that do not correspond with those in the main manuscript.

**Questions:**

See the weaknesses.

**Limitations:**

Recent years have seen the emergence of various tensor-based multi-view clustering methods and anchor-based approaches. It is crucial to compare and contrast these methods in the manuscript to highlight their fundamental differences.

---

> ### Author Rebuttal · Authors · 2024-08-06
>
> Thank you for your thoughtful and detailed review of our manuscript. We have carefully considered each of your comments and addressed them point by point below:
>
> **Weakness 1:** The introduction of Enhanced Anchor Dictionary (EAD) lacks clarity. Could the authors clarify the fundamental differences between EAD, traditional dictionary learning, and anchor dictionary learning? Additionally, how does the methodology incorporate hidden data?
>
> **A1:** The fundamental difference between traditional dictionary learning, anchor dictionary learning, and our Enhanced Anchor Dictionary (EAD) lies in the dictionary representation used. Traditional dictionary learning employs the observed data itself to recover a subspace representation of size $n \times n$. Anchor dictionary learning, on the other hand, utilizes a subset of samples from the observed data as anchor points to recover a subspace representation of size $n \times l$. In contrast, while EAD also uses anchor points, it additionally accounts for the influence of hidden data to recover a subspace representation of size $n \times l$. This approach not only reduces computational complexity but also addresses the imprecision arising from insufficient sampling in observed data, resulting in a more accurate and efficient representation.
>
> **Weakness 2:** In Section 4.3, the authors reference Figure 11 in the first part, but the figure numbering is incorrect and needs to be corrected.
>
> **A2:** Thank you for pointing out the discrepancy with the figure numbering. We will correct the reference to Figure 11 in Section 4.3 to ensure it aligns with the correct figure numbering in the manuscript.
>
> **Weakness 3:** From Table 2 in the manuscript, it appears that most methods have a standard deviation of 0.000. Could the authors explain why these methods exhibit no standard deviation?
>
> **Q3:** Indeed, most methods, including our STONE algorithm, show a standard deviation of 0.000 across all datasets. For comparison methods like SFMC, GMC, MVCtopl, and TBGL, this is due to their clustering strategies which impose connectivity constraints on consensus graphs, ensuring that connected components accurately reflect the true clustering labels and thereby avoiding the variability inherent in spectral clustering. In the case of our STONE algorithm, the stability observed with k-means clustering is attributed to applying k-means to the left singular vector of the concatenated matrix $\bar {\mathbf Z} = \frac{1}{\sqrt m} [\mathbf Z^1,...,\mathbf Z^m]$, which is already well-separated and stable, leading to consistent clustering results across multiple runs. We will explore this issue further in the revised manuscript.
>
>  **Weakness 4:** Based on Figure 10, why does the performance of the proposed method not exhibit a linear positive correlation with the number of anchor points? Typically, increasing the number of anchor points should lead to acquiring more valuable information.
>
> **A4:** Thank you for your insightful question about the performance of our method in relation to the number of anchor points, as illustrated in Figure 10. The lack of a linear positive correlation between performance and the number of anchor points can be attributed to diminishing returns, where the incremental benefit of additional anchor points decreases beyond a certain threshold. Additionally, an excessive number of anchor points can introduce redundancy and noise, potentially degrading performance. Moreover, the quality of anchor points is crucial; additional points that lack discriminative power may not improve, and could even harm, performance. We will elaborate on these factors in the revised manuscript to clarify the observed trends.
>
> **Weakness 5:** The pseudocode displayed for Algorithm 1 in the appendix has formula indices that do not correspond with those in the main manuscript.
>
> **A5:** Thank you for pointing out the discrepancy between the formula indices in the pseudocode for Algorithm 1 in the appendix and those in the main manuscript. We will correct the indices in the pseudocode to ensure they match those referenced in the main text.
>
> **Limitation 1:** Recent years have seen the emergence of various tensor-based multi-view clustering methods and anchor-based approaches. It is crucial to compare and contrast these methods in the manuscript to highlight their fundamental differences.
>
> **A6:** Thank you for your constructive feedback. We will incorporate a comprehensive comparison and discussion of recent tensor-based multi-view clustering methods, anchor-based approaches, and our STONE method in the revised manuscript. This will help to clearly highlight the fundamental differences and provide a thorough understanding of each approach.

---

> > ### Comment · Reviewer_iXTf · 2024-08-12
> >
> > Thanks for your responses, and my questions have been resolved.

---

> > > ### Author Response · Authors · 2024-08-12
> > >
> > > We sincerely appreciate the time and effort you dedicated to reviewing our work and for providing constructive feedback.

---

### Official Review · Reviewer_QZiQ · 2024-07-10

**Soundness:** 3
**Presentation:** 3
**Contribution:** 2
**Rating:** 4
**Confidence:** 4

**Summary:**

The paper presents a novel framework, STONE (Scalable TMSC framework with Triple information Enhancement), addressing significant limitations in current Tensor-based Multi-view Subspace Clustering (TMSC) methods. The proposed approach aims to reduce computational complexity, improve subspace representation accuracy, and better handle noise-related singular values in tensor data. Through an enhanced anchor dictionary learning mechanism, an anchor hypergraph Laplacian regularizer, and the use of an improved hyperbolic tangent function, the authors demonstrate superior performance compared to state-of-the-art (SOTA) methods.

**Strengths:**

1.By incorporating this regularizer, the method preserves the inherent geometric structure of the data, leading to more accurate subspace representation.
2. Extensive experimentation on a variety of datasets demonstrates the method's effectiveness and efficiency, surpassing SOTA methods.
3. The framework's design allows for scalable application to large datasets, making it practical for real-world scenarios with high-dimensional multi-view data.

**Weaknesses:**

1. In Figure 5, the authors should carefully check the name of each subfigure.
2. The authors state that A represents l anchors of the v-th view, with orthogonal constraints for optimal discriminability. However, the matrix A is d*l, and the size of constraint A * A’ is d * d， It is not true that features have orthogonality to each other.
3. In Equation 7, the LWF constraint introduces the weighted coefficient vector to balance the weight of each view, while this coefficient vector is not seen in Equation 10. Thu authors should check it.
4. The trace of Z in Equation 8 should be added.
5. In the experiments, specifications need to be given as to what number of anchors is selected.
6. Although the performance of the proposed method is superior to other compared methods, the number of parameters is too many, including four parameters and a parameter number of anchors. According to the code, we can obtain 8 * 8 * 8 * 8 * 6=24576 combinations of parameters. Authors should optimize the model, not simply add constraints together.
7. The authors should explain the difference between the proposed model and the existing work ''Anchor Structure Regularization Induced Multi-view Subspace Clustering via Enhanced Tensor Rank Minimization'' published in ICCV.

**Questions:**

None

---

> ### Author Rebuttal · Authors · 2024-08-06
>
> We sincerely appreciate your detailed review of our manuscript. We have carefully considered all your comments and provide our responses to each point below:
>
> **Weakness 1:** In Figure 5, the authors should carefully check the name of each subfigure.
>
> **A1:** We have carefully checked the names of the subfigures in Figure 5 and can confirm that they are correct. Each subfigure in Figure 5 pertains to the HW dataset and illustrates the cross-validation of three parameters: $\alpha$, $\beta$ and $\gamma$.
>
> **Weakness 2:** Typo in Orthogonality Constraint for $\mathbf A \mathbf A^T = \mathbf I$
>
> **A2:** Thank you for pointing out the typo. The correct constraint should indeed be $\mathbf A^T \mathbf A = \mathbf I$ instead of $\mathbf A \mathbf A^T = \mathbf I$. We will revise the manuscript to accurately reflect this correction and provide a clear explanation of how this constraint impacts feature representations.
>
> **Weakness 3:** Why is the weighted coefficient vector from Eq. (7) not present in Eq. (10)?
>
> **A3:** Each element of weighted coefficient vector in Eq. (7) is indeed set to 1, which is why it is not explicitly shown in Eq. (10). This simplification is discussed in the explanation following Eq. (7) in the manuscript. In the revised manuscript, we will provide a more detailed explanation to clarify this point.
>
> **Weakness 4:** The trace of $\mathbf Z$ in Eq. (8) should be added.
>
> **A4:** Thank you for your insightful review. It appears that there might be a minor misunderstanding regarding Eq. (8). We believe the issue may concern the missing transpose of $\mathbf Z$ rather than the trace. We will thoroughly reexamine Eq. (8) to ensure that the transpose is correctly included and make the necessary revisions.
>
> **Weakness 5**: Specifications Needed for Number of Anchors in Experiments.
>
> **A5:** In our experiments, the number of anchors for the Scene15 dataset was set to 2c, while for the other datasets, it was set to c. This is discussed on lines 667-669 of the manuscript. We will provide a more detailed discussion of these settings in the revised manuscript to ensure clarity and completeness.
>
> **Weakness 6:** Although the performance of the proposed method is superior to other compared methods, the number of parameters is too many, including four parameters and a parameter number of anchors. According to the code, we can obtain 8 * 8 * 8 * 8 * 6=24576 combinations of parameters. Authors should optimize the model, not simply add constraints together.
>
> **A6:** Our model indeed involves 5 parameters: 2 built-in parameters ($\delta$ and $l$) and 3 balancing parameters ($\alpha$, $\beta$, and $\gamma$). For clarity, the source code includes all theoretically possible combinations, but in practice, the number of parameter configurations we explore is much smaller. Specifically, $\delta$ and $l$ are tuned independently, and we perform cross-validation only for the balancing parameters. $\delta$ and $l$ are adjusted within the ranges [0.1, 0.5, 1, 1.5, 2, 5] and [1c, 7c], respectively, resulting in a total of 13 combinations. The balancing parameters, $\alpha$, $\beta$, and $\gamma$, are tuned using cross-validation within the range [1e-5, 1e1], leading to 343 combinations (7 × 7 × 7). Thus, the total number of parameter configurations we actually explore is 13 + 343 = 356, which is significantly fewer than the theoretical count of 24,576 combinations (as detailed in the experimental section of the manuscript). Furthermore, as shown in the experimental analysis, when $\delta$ and $l$ are fixed at $\delta = 1$ and $l = 2c$, our model achieves optimal and stable performance across all datasets. This indicates that these parameters, even when set to these default values, still provide optimal performance, further reducing the parameter burden.
>
> Overall, we only need to fine-tune the three balancing parameters, which is typical for many SOTA methods. Additionally, our model innovatively integrates EAD, HTR, and AHR into a unified framework, with each component demonstrating significant effectiveness and originality in multi-view clustering.
>
> **Weakness 7:** Difference between the proposed model and the existing work ''ASR-ETR''  in ICCV.
>
> **A7:** ASR-ETR [30] is a highly representative contribution to the multi-view clustering field. There are three fundamental differences between our STONE model and ASR-ETR:
>
> **(1). Dictionary Learning Strategy:** The ASR-ETR model relies on traditional self-representation learning, which recovers subspace representations based on the original anchor representations. It does not address the inaccuracies caused by insufficient sampling of the original anchors. In contrast, our STONE model uses an enhanced dictionary learning strategy that more effectively captures and mitigates the inaccuracies due to undersampling, thereby improving the precision of subspace representations.
>
> **(2). Manifold Regularization:** The ASR-ETR model employs traditional Laplacian manifold regularization, which captures only pairwise manifold relationships. Our STONE model, on the other hand, utilizes hypergraph Laplacian manifold regularization, which captures both pairwise linear relationships and higher-order nonlinear topological relationships among multiple data points.
>
> **(3). Tensor Rank Constraint:** While ASR-ETR applies an Enhanced Tensor Rank constraint for low-rank tensor representation, our STONE model introduces a novel Hyperbolic Tangent Rank constraint. This new constraint is designed to capture more nuanced differences between singular values in tensor data, enhancing robustness and representation accuracy.
>
> Overall, our work builds upon the foundation laid by ASR-ETR, pushing the boundaries of multi-view clustering through these novel contributions in dictionary representation, subspace representation, and tensor representation. We appreciate the opportunity to clarify these distinctions and will provide a more detailed comparison in the revised manuscript.

---

> ### Author Response · Authors · 2024-08-14
>
> Dear Reviewer QZiQ,
>
> Thank you very much for your insightful feedback. We have carefully responded to all the concerns you raised about our manuscript. With the rebuttal deadline approaching, we would like to gently inquire if you have any further questions or need additional information. If not, we would greatly appreciate it if you could revisit your score.
>
> Thank you for your time and thoughtful consideration.
>
> Best regards,
>
> Author

---

### Official Review · Reviewer_6B6d · 2024-07-12

**Soundness:** 4
**Presentation:** 4
**Contribution:** 3
**Rating:** 7
**Confidence:** 5

**Summary:**

The authors introduce the STONE framework, a Tensor-based Multi-view Subspace Clustering (TMSC) approach, designed to surmount the paramount limitations inherent in contemporary methodologies. By augmenting anchor dictionary learning, they adeptly reconstruct low-rank structures, resulting in a reduction in computational intricacy and bolstering resilience, particularly in scenarios constrained by limited dictionaries. Furthermore, the framework incorporates an innovative anchor hypergraph Laplacian regularizer, ensuring the preservation of data geometry within subspace representations, and leverages an enhanced hyperbolic tangent function for precision in tensor rank approximation.

**Strengths:**

1)The motivation of this paper is explicit. The authors strive to elevate cluster performance while minimizing computational complexity by refining anchor dictionary learning, incorporating anchor hypergraph Laplacian regularization, and utilizing an enhanced hyperbolic tangent function for precise rank approximation.
2)The proposed method stands out from other state-of-the-art (SOTA) techniques, surpassing their effectiveness and efficiency. Notably, the authors have graciously shared the source code, facilitating further exploration and validation of their innovative approach.
3)The authors' rigorous theoretical analyses, encompassing computational complexity and convergence evaluations, provide a solid foundation for the validity and reliability of their method, further strengthening the overall impact of their research.

**Weaknesses:**

1) The authors clarify that their proposed method employs an advanced dictionary learning mechanism to delve into and uncover latent data, which refers to information that is inherently present but not directly observable or accessible without specific techniques. However, the exact nature of this "hidden data" could be further elaborated to avoid ambiguity. It is crucial to distinguish this concept from missing multi-view data, which pertains to instances where certain views or features of the data are absent or incomplete. The enhanced dictionary learning aims to capture and represent these underlying, yet hidden, patterns and structures within the data, distinct from simply filling in gaps caused by missing views.
2) There are some typographical errors in the manuscript, such as on line 584 where "Eq. (11)" seems to have an incorrect sequence number.

**Questions:**

1) Does the dimensionality k of the latent data remain consistent across various views?
2) The proposed method innovatively employs hypergraph Laplacian regularization to delve into and uncover local intrinsic manifold structures within the data. How does this hypergraph differ from traditional anchor graphs? Additionally, how many anchor points are set in the hypergraph, and is this consistent across all datasets?
3) The method introduces multiple variables, each requiring careful initialization to ensure the stability and convergence of the optimization process. The authors should clarify the initialization strategies adopted for each variable. How are each of these variables initialized?
4) In Section Experimental Setup, how is $\delta$ set across different datasets?

**Limitations:**

See the weaknesses.

---

> ### Author Rebuttal · Authors · 2024-08-06
>
> Thank you for your thoughtful review of our manuscript. We have carefully considered each of your comments and addressed them point by point below:
>
> **Weakness 1:**  The authors clarify that their proposed method employs an advanced dictionary learning mechanism to delve into and uncover latent data, which refers to information that is inherently present but not directly observable or accessible without specific techniques. However, the exact nature of this "hidden data" could be further elaborated to avoid ambiguity. It is crucial to distinguish this concept from missing multi-view data, which pertains to instances where certain views or features of the data are absent or incomplete. The enhanced dictionary learning aims to capture and represent these underlying, yet hidden, patterns and structures within the data, distinct from simply filling in gaps caused by missing views.
>
> **A1:** In our study, "hidden data" refers to latent structures and patterns within the dataset that are not directly observable. This is different from missing multi-view data, which involves cases where certain views or features are absent or incomplete. Our advanced dictionary learning mechanism aims to reveal these underlying patterns and structures rather than simply addressing gaps from missing views. We appreciate your suggestion for further clarification and will include a more detailed explanation of "hidden data" in the revised manuscript.
>
> **Weakness 2:** There are some typographical errors in the manuscript, such as on line 584 where "Eq. (11)" seems to have an incorrect sequence number.
>
> **A2:** Thank you for pointing out the typographical errors. We will correct the sequence number for "Eq. (11)" on line 584, along with any other errors identified in the manuscript.
>
> **Question 1:** Does the dimensionality k of the latent data remain consistent across various views?
>
> **A3:** The latent data in our model is intended to address the limitations in the original data due to insufficient sampling, serving as an idealized supplement across different views. Theoretically, the latent data could differ between views. However, in our approach, we use skinny SVD theory to model the influence of latent data as a regularization term. We do not explicitly recover the latent data itself. Therefore, the dimensionality k we refer to is symbolic and represents the latent data dimensions rather than an actual recovered value.
>
> **Question 2:** The proposed method innovatively employs hypergraph Laplacian regularization to delve into and uncover local intrinsic manifold structures within the data. How does this hypergraph differ from traditional anchor graphs? Additionally, how many anchor points are set in the hypergraph, and is this consistent across all datasets?
>
> **A4:** **Difference from Traditional Anchor Graphs:** The hypergraph Laplacian regularization used in our method differs from traditional anchor graphs primarily in its ability to capture higher-order relationships among data points. While traditional anchor graphs focus on pairwise relationships between data points, hypergraphs extend this concept to capture relationships among groups of points. This allows our method to uncover more complex local intrinsic manifold structures within the data by considering interactions among multiple points simultaneously. **The number of anchor points** is set to 3, and this value is consistent across all datasets.
>
> **Question 3**: The method introduces multiple variables, each requiring careful initialization to ensure the stability and convergence of the optimization process. The authors should clarify the initialization strategies adopted for each variable. How are each of these variables initialized?
>
> **A5:** In our method, all variables are initialized as zero matrices. This initialization approach is outlined in the pseudocode available in the supplementary materials (see page 16). We will provide a more detailed discussion of this initialization strategy in the revised manuscript.
>
> **Question 4:** In Section Experimental Setup, how δ set across different datasets?
>
> **A6:** Thank you for your question. Initially, we set the value of δ empirically to 1. We then fine-tuned this parameter to determine the optimal values for each dataset. Specifically, δ was set to 1 for the datasets NGs, BBCSport, HW, Scene15, MSRCV1, and ALOI-100. For the datasets Caltech101-all and CIFAR10, we adjusted δ to 0.5 and 0.1, respectively. We will provide a more detailed explanation of these parameter settings in the revised manuscript to ensure greater clarity.

---

> > ### Comment · Reviewer_6B6d · 2024-08-14
> >
> > Thanks for the authors' response. My concerns have been addressed.

---

> > > ### Author Response · Authors · 2024-08-14
> > >
> > > Thank you for your response. We are pleased to hear that our rebuttal has resolved your concerns. We appreciate your valuable feedback and the time you have spent reviewing our manuscript, which has positively contributed to its improvement.
> > >
> > > Best regards,
> > >
> > > Author

---

### Decision · Program_Chairs · 2024-09-25

**Decision:**

Accept (poster)

**Comment:**

The paper is about multi-view subspace clustering, a fundamental and significant problem to study. While the review scores diverge greatly, no critical concerns can be found in the reviews. Reviewer QZiQ and Reviewer eQ9V, who give respectively scores of 4 and 3, only complain minor issues such as typos, which are not difficult to fix. I have also read the paper quickly, and my assertion is the paper contains solid contribution that would be valuable to this conference. I would recommend accepting the paper.